# Drosophila TRIM32 cooperates with glycolytic enzymes to promote cell growth

Simranjot Bawa[1], David S Brooks[1], Kathryn E Neville[2], Marla Tipping[2], Md Abdul Sagar[3], Joseph A Kollhoff[1], Geetanjali Chawla[4,5], Brian V Geisbrecht[1], Jason M Tennessen[5], Kevin W Eliceiri[3], Erika R Geisbrecht[1]*

[1]Department of Biochemistry and Molecular Biophysics, Kansas State University, Manhattan, United States; [2]Department of Biology, Providence College, Providence, United States; [3]Laboratory for Optical and Computational Instrumentation, Department of Biomedical Engineering, University of Wisconsin-Madison, Madison, United States; [4]Regional Centre for Biotechnology, NCR Biotech Science Cluster, 3rd Milestone, Faridabad-Gurgaon Expressway, Faridabad, India; [5]Department of Biology, Indiana University, Bloomington, United States

**Abstract** Cell growth and/or proliferation may require the reprogramming of metabolic pathways, whereby a switch from oxidative to glycolytic metabolism diverts glycolytic intermediates towards anabolic pathways. Herein, we identify a novel role for TRIM32 in the maintenance of glycolytic flux mediated by biochemical interactions with the glycolytic enzymes Aldolase and Phosphoglycerate mutase. Loss of Drosophila TRIM32, encoded by thin (tn), shows reduced levels of glycolytic intermediates and amino acids. This altered metabolic profile correlates with a reduction in the size of glycolytic larval muscle and brain tissue. Consistent with a role for metabolic intermediates in glycolysis-driven biomass production, dietary amino acid supplementation in tn mutants improves muscle mass. Remarkably, TRIM32 is also required for ectopic growth - loss of TRIM32 in a wing disc-associated tumor model reduces glycolytic metabolism and restricts growth. Overall, our results reveal a novel role for TRIM32 for controlling glycolysis in the context of both normal development and tumor growth.

*For correspondence: geisbrechte@ksu.edu

Competing interests: The authors declare that no competing interests exist.

## Introduction

The metabolism of all cells must adapt to meet the energetic and biosynthetic needs of growth and homeostasis (*Lloyd, 2013*; *Zhu and Thompson, 2019*). For example, tissues composed of non-dividing, differentiated cells must strike a balance between catabolic pathways that provide energy for cellular homeostasis and anabolic pathways that repair the cell and generate cell-type-specific molecules (*Lloyd, 2013*). In contrast, the metabolic requirements of cell growth and proliferation often require a shift toward anabolic pathways that favors the synthesis of macromolecules, such as proteins, lipids, nucleic acids, and complex carbohydrates (*Zhu and Thompson, 2019*). Striking this delicate balance between degradative and biosynthetic processes requires the integration of extracellular and intracellular information by complex signaling networks.

The mechanisms by which cell proliferation and tissue growth rewire metabolism to enhance biosynthesis are diverse and complex (*Lunt and Vander Heiden, 2011*). These changes in metabolic flux involve pathways such as the pyrimidine and purine biosynthesis, one carbon metabolism, and the interplay between the citric acid cycle and amino acids pools. However, the pathway most commonly associated with enhanced biosynthesis is glycolysis, where in biological systems ranging from yeast to human T-cells, glycolytic flux is often elevated in the context of cell growth and proliferation

(*Zhu and Thompson, 2019*). This observation is particularly apparent in the fruit fly *Drosophila melanogaster*, where the onset of larval development is preceded by a metabolic switch that induces the coordinate upregulation of genes involved in glycolysis, the pentose phosphate pathway, and lactate dehydrogenase (LDH) (*Tennessen et al., 2011*). The resulting metabolic program allows larvae to use dietary carbohydrates for both energy production and biomass accumulation. Moreover, studies of *Drosophila* larval muscles reveal that this metabolic transition is essential for muscle growth and development, suggesting that glycolysis serves a key role in controlling growth (*Tennessen et al., 2014b*). The mechanisms that control glycolysis specifically in larval muscle, however, remain relatively unexplored. As a result, *Drosophila* larval development provides an excellent model for understanding how glycolysis and biomass production are regulated in a rapidly growing tissue. Moreover, since larval muscle increases in size without cell divisions, larval muscle provides an unusual opportunity to understand how glycolytic metabolism promotes growth independent of cell division.

Of the known factors that promote muscle development, TRIM32 is an intriguing candidate for coordinating metabolism with cell growth. This protein is a member of the Tripartite motif (TRIM)-containing family of proteins defined by an N-terminal RING domain, one or two B-boxes, a coiled-coil domain, and a variable C-terminal region (*Tocchini and Ciosk, 2015*; *Watanabe and Hatakeyama, 2017*). In TRIM32, six N̲cl-1, H̲T2A, L̲in-41 (NHL) repeats comprise the C-terminus and are proposed to mediate the diverse functions of TRIM32, including cell proliferation, neuronal differentiation, muscle physiology and regeneration, and tumorigenesis (*Lazzari and Meroni, 2016*; *Tocchini and Ciosk, 2015*; *Watanabe and Hatakeyama, 2017*). A single mutation in the B-box region of TRIM32 causes the multisystemic disorder Bardet-Biedl syndrome (BBS) (*Chiang et al., 2006*), while multiple mutations that cluster in the NHL domains result in the muscle disorders Limb-girdle muscular dystrophy type 2H (LGMD2H) and Sarcotubular Myopathy (STM) (*Borg et al., 2009*; *Frosk et al., 2005*; *Lazzari et al., 2019*; *Nectoux et al., 2015*; *Neri et al., 2013*; *Schoser et al., 2005*; *Servián-Morilla et al., 2019*).

A complete understanding of TRIM32 function is confounded by its ubiquitous expression and multitude of potential substrates for E3 ligase activity via the RING domain. Many known TRIM32 target substrates include proteins implicated in muscle physiology (*Albor et al., 2006*; *Cohen et al., 2014*; *Cohen et al., 2012*; *Kudryashova et al., 2005*; *Locke et al., 2009*; *Volodin et al., 2017*) or the prevention of satellite cell senescence (*Kudryashova et al., 2012*; *Mokhonova et al., 2015*; *Servián-Morilla et al., 2019*), consistent with a role for TRIM32 in LGMD2H. However, additional polyubiquitinated substrates, including p53, Abi2, Piasy, XIAP, and MYCN, are implicated in tumorigenesis (*Albor et al., 2006*; *Izumi and Kaneko, 2014*; *Kano et al., 2008*; *Liu et al., 2014*; *Ryu et al., 2011*). Importantly, TRIM32 protein levels are upregulated in multiple tumor types, suggesting that TRIM32 is a key player in growth regulation (*Horn et al., 2004*; *Ito et al., 2017*; *Zhao et al., 2018*). There is precedence for NHL function in controlling cell proliferation as two other *Drosophila* NHL-containing proteins, Brat and Mei-P26, act as tumor suppressors in the larval brain and female germline, respectively (*Arama et al., 2000*; *Edwards et al., 2003*).

Here, we provide a novel mechanism for TRIM32 in cell growth. Our data show that TRIM32 promotes glucose metabolism through the stabilization of glycolytic enzyme levels. This increased rate of TRIM32-mediated glycolytic flux generates precursors that are utilized for biomass production. Surprisingly, this mechanism operates in both non-dividing muscle cells as well as in proliferating larval brain cells, demonstrating a universal metabolic function for TRIM32 in growth control.

## Results

### TRIM32 binds to glycolytic enzymes

While NHL domain-containing proteins can interact with both RNAs and proteins (*Tocchini and Ciosk, 2015*; *Watanabe and Hatakeyama, 2017*), few bona fide TRIM32 binding partners have been identified. Causative mutations in human LGMD2H cluster in the NHL repeats (*Figure 1A*; *Borg et al., 2009*; *Cossée et al., 2009*; *Frosk et al., 2005*; *Lazzari et al., 2019*; *Nectoux et al., 2015*; *Neri et al., 2013*; *Schoser et al., 2005*), suggesting that this region may mediate protein-protein interactions important in disease prevention. We previously showed that mutations in *Drosophila* TRIM32 also show progressive larval muscle degeneration, despite modest

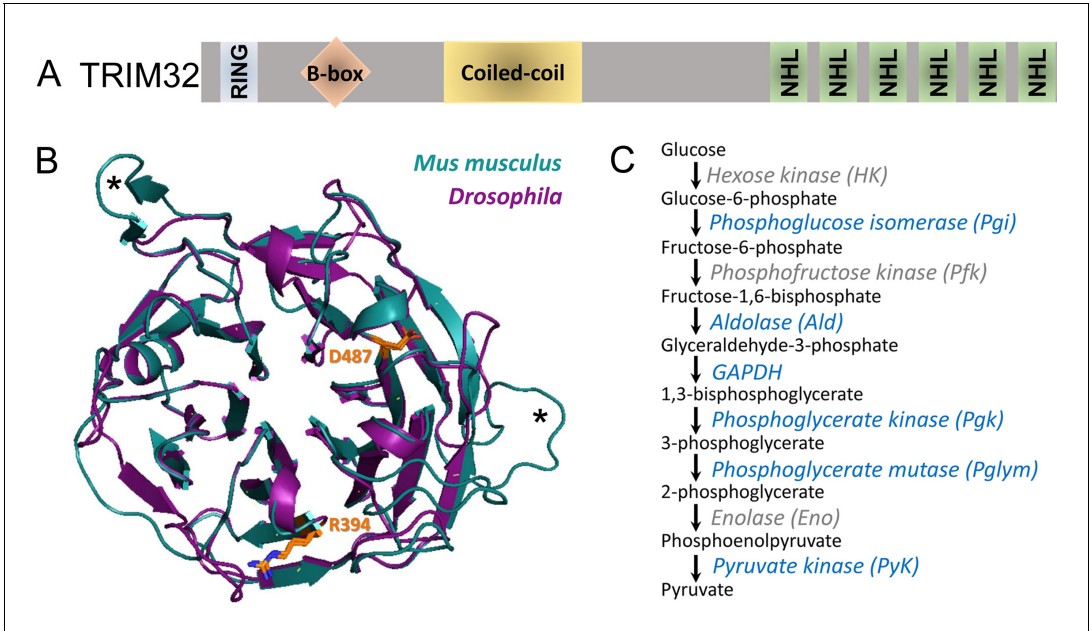

**Figure 1.** The NHL region of *Drosophila* TRIM32 is structurally conserved. (**A**) Schematic showing the RING, B-box, coiled-coil, and NHL domains in TRIM32. (**B**) Superimposed protein structures of the six NHL repeats in *Drosophila* (magenta) and *Mus musculus* (blue) TRIM32. Each NHL repeat consists of four antiparallel beta sheets that are arranged toroidally around a central axis. Mouse NHL has two additional loops (asterisks) not present in the fly protein. The positions and orientation of both R394/R1114 and D487/D1211 are identical between *Mus musculus* and *Drosophila* (orange). (**C**) The glycolytic pathway. Peptides corresponding to enzymes that co-purified with TRIM32_NHL are shown in blue.

The online version of this article includes the following source data and figure supplement(s) for figure 1:

**Source data 1.** X-ray diffraction data collection and refinement statistics.
**Source data 2.** Proteins identified via MS that co-purify with TRIM32-NHL Experiment #1 (1418) Experiment #2 (1426.1) Experiment #3 (1426.2).
**Figure supplement 1.** Crystal structure of the *Drosophila* TRIM32_NHL domain.

sequence identity (~42%) across the NHL domains (*LaBeau-DiMenna et al., 2012*). To gain molecular-level insight into the functional conservation between the NHL regions in fly and mammalian TRIM32, we obtained a crystal structure of TRIM32_NHL at 2.6 Å resolution (Rwork/Rfree = 18.2/22.8%) (*Figure 1—figure supplement 1A–C*; *Figure 1—source data 1*). The structure was solved by molecular replacement, using the NHL-repeat region of the *Drosophila* Brat protein as a search model (PDB Code 1Q7F) (*Edwards et al., 2003*). This structure features a six-bladed β-propeller, where each NHL repeat is comprised of four antiparallel β-strands (*Figure 1B*; *Figure 1—figure supplement 1C*). We then used the TRIM32_NHL structure to construct a model for mouse TRIM32_NHL using SWISS-MODEL. Indeed, the superimposition of these two structures demonstrate that the NHL region of *Drosophila* TRIM32 is a faithful model to understand NHL function (*Figure 1B*).

To identify TRIM32_NHL-interacting proteins, we performed in vivo pulldowns followed by mass spectrometry (MS) to uncover associated peptides. Briefly, the C-terminal region of *Drosophila* TRIM32 containing all six NHL domains (AA 1061–1353) was fused in-frame with an N-terminal Glutathione S-transferase (GST) tag. This GST_TRIM32_NHL protein was expressed, purified, and incubated with third instar larval (L3) lysates. These resulting protein complexes were subjected to MS to detect peptide fragments that co-purify with TRIM32_NHL. In addition to the recovery of our bait protein, we detected peptides corresponding to Tropomyosin (Tm) and Troponin T (TnT) (*Figure 1—source data 2*), which are both known polyubiquitinated substrates of mammalian TRIM32 (*Cohen et al., 2012*). Surprisingly, glycolytic enzymes were also enriched in the list of possible NHL-binding proteins (*Figure 1C*; *Figure 1—source data 2*) and were further evaluated for physical interactions with TRIM32.

We utilized two independent methods to validate candidate protein interactions with TRIM32. First, an in vitro binding assay using purified proteins was developed with Tropomyosin 2 (Tm2) as a

positive control. Indeed, incubation of TRIM32_NHL together with His-tagged Tm2 confirmed a physical interaction between these two proteins (*Figure 2—figure supplement 1A*). This assay was then used to test for physical interactions with candidate glycolytic enzymes. To this end, we expressed and purified His-tagged Aldolase (Ald) or His-tagged Phosphoglycerate mutase 78 (Pglym), both of which were low or absent in control pulldowns. Both Ald and Pglym were found to directly interact with the NHL domains of TRIM32 (*Figure 2A,B*). Note that no binding was detected between TRIM32_NHL and the His-tagged control protein SCIN, ruling out non-specific binding of the His tag. To extend these findings, we utilized immunoprecipitation experiments to test for in vivo interactions with full-length TRIM32. As expected, immunoprecipitation of TRIM32 successfully pulled-down Tm as an interacting protein at the predicted molecular weight of ~37 kD (*Figure 2—figure supplement 1B*). We also confirmed that both Ald and Pglym co-immunoprecipitate with anti-TRIM32 antibody. Interestingly, the bands corresponding to these proteins migrated at a higher molecular weight than the expected size present in input lysates (*Figure 2C,D*; asterisk), suggestive of a post-translational modification. Two approaches were taken to confirm the specificity of the slower migrating bands that correspond to Ald or Pglym. First, we concentrated L3 lysates before SDS-PAGE analysis. After overexposure of the western blots, faint bands corresponding to the higher molecular weight forms of Ald or Pglym were present (*Figure 2C,D*; asterisk). We also immunoprecipitated TRIM32 from larval lysates and probed for Ald or Pglym using antibodies raised against the orthologous human proteins (ALD or PGAM1). While these antibodies did not cross react well in input lysates, there was an obvious band corresponding to the higher molecular weight forms of both Ald and Pglym after pulling down TRIM32 (*Figure 2—figure supplement 1D,E*). Collectively, these data demonstrate that Ald and Pglym interact with the NHL domain of TRIM32 and in vivo, a subpopulation of these post-translationally modified glycolytic enzymes can be found in a complex with TRIM32.

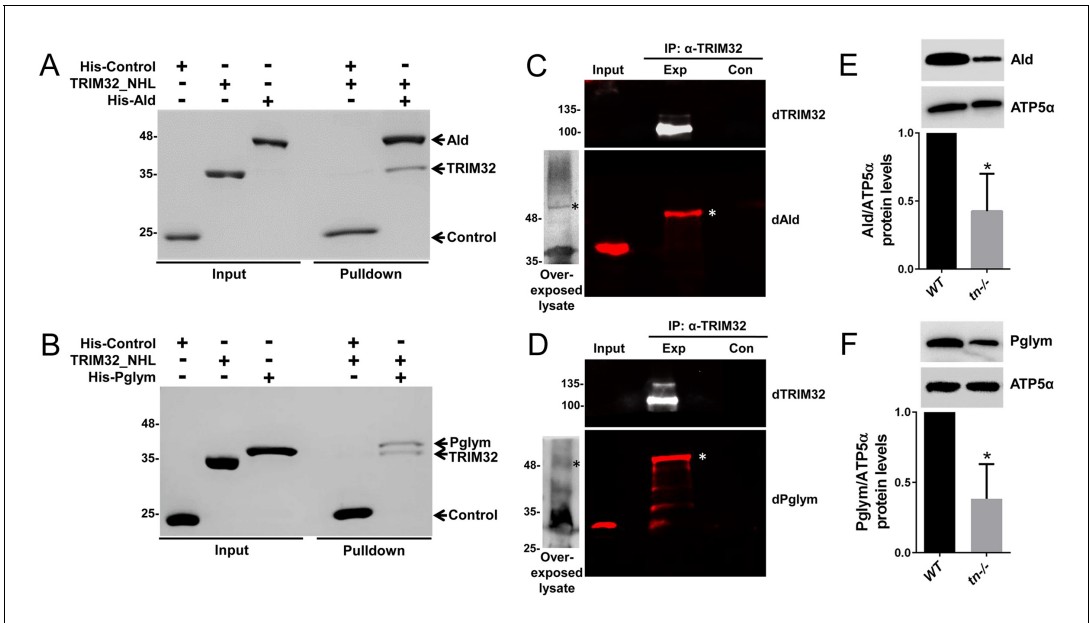

**Figure 2.** *Drosophila* TRIM32 physically interacts with the glycolytic enzymes Ald and Pglym78. (**A,B**) In vitro binding assays. Untagged TRIM32_NHL was incubated with either the His-tagged SCIN control protein or the His-tagged candidate proteins Ald (**A**) or Pglym (**B**). After washing in 300 mM NaCl and 0.1% Triton, each of these protein complexes was separated by SDS-PAGE followed by Coomassie staining. Ald and Pglym proteins directly bind the NHL region of TRIM32, while no interaction with the His-SCIN control protein is observed. (**C,D**) Western blotting with antibodies against *Drosophila* Ald (dAld; **C**) or *Drosophila* Pglym (dPglym; **D**) detects higher molecular weight bands (asterisk) upon immunoprecipitation of *Drosophila* TRIM32, but not in control lanes. The observed molecular weights of Ald or Pglym in input larval lysates (~30 µg) is predominant over a higher migrating form that can be visualized after overexposure of blots with concentrated lysate (~300 µg). (**E,F**) Western blots showing that mutations in *tn* reduce Ald (**E**) or Pglym (**F**) protein levels ~ 50% quantitated relative to ATP5α in L3 larvae. N = 3. Mean +/- SD (*, p<0.05).
The online version of this article includes the following figure supplement(s) for figure 2:

**Figure supplement 1.** TRIM32 co-purifies with Tm2 and other proteins.

The majority of TRIM32 studies have focused on the biological impact of polyUb chain addition onto substrates (*Cohen et al., 2012*; *Kudryashova et al., 2005*; *Liu et al., 2014*; *Locke et al., 2009*; *Mokhonova et al., 2015*). This modification targets a protein for proteasomal degradation and usually increases protein substrate levels in the absence of its E3 counterpart. Indeed, the known mammalian substrate Tm binds to TRIM32_NHL (*Cohen et al., 2012*) and accordingly, *Drosophila* Tm was modestly increased in larvae mutant for *tn* (*Figure 2—figure supplement 1C*). Strikingly, this same loss of TRIM32 resulted in a ~ 50% decrease in the protein levels of Ald or Pglym78 (*Figure 2E,F*). This downregulation was not due to a decrease in TRIM32-mediated transcription as mRNA levels of *Ald* or *Pglym* were not altered in *tn-/-* (*Figure 2—figure supplement 1F,G*). Taken together, these data highlight a unique role for TRIM32 in the stabilization of glycolytic enzyme levels.

## Loss of TRIM32 disrupts glycolytic metabolism in larval muscles

To understand the biological implications for the physical interaction between TRIM32 and the glycolytic enzymes Ald and Pglym, we profiled 85 metabolites present in *WT* or *tn-/-* L3 larvae. Principal component analysis (PCA) revealed a distinct separation between *WT* and *tn-/-* experimental groups (*Figure 3—figure supplement 1A*). Examination of individual metabolites upon loss of TRIM32 showed a decrease in the terminal glycolytic products pyruvate and lactate as well as significant depletion of the glucose-derived metabolites glycerol-3-phosphate and 2-hydroxyglutarate (*Figure 3A,B*). These metabolic changes suggest that TRIM32 is required for sustaining the conversion of glucose to pyruvate and lactate in L3 larvae.

The production of pyruvate at the end of glycolysis has two fates, either a reduction to lactate or oxidation to $CO_2$ in the mitochondrion (*Mookerjee et al., 2015*; *TeSlaa and Teitell, 2014*). To determine if reduced lactate levels in *tn-/-* results in increased respiration and therefore elevated ATP synthesis, we assessed these two outputs of overall energy metabolism in control or mutant whole larvae. Respirometry analysis revealed a modest increase in $CO_2$ production upon loss of TRIM32 (*Figure 3—figure supplement 1B*). However, ATP levels were significantly decreased in *tn-/-* larvae (*Figure 3—figure supplement 1C*). Given that different larval tissues may have altered metabolic profiles that influence outputs of whole body metabolism, we sought to assess glycolytic activity in individual tissues.

A unique aspect of *Drosophila* larval development is the 200-fold increase in body size from first instar (L1) larvae through the L3 stage. In order to support high levels of biomass accumulation, glycolytic metabolism is increased prior to larval growth (*Tennessen et al., 2011*; *Tennessen et al., 2014b*). One larval tissue that requires high glycolytic activity is the somatic musculature, where *WT* body wall muscles undergo dramatic growth as development proceeds from second instar larvae (L2) (*Figure 3C,G*) to the L3 stage (*Figure 3D,G*). We, and others, previously reported a loss of structural integrity in TRIM32-deficient muscles (*Domsch et al., 2013*; *LaBeau-DiMenna et al., 2012*) and assumed that the associated reduction in muscle size was a secondary consequence of progressive tissue degeneration. While multiple independent alleles of *tn* have been shown to produce smaller muscles (*Domsch et al., 2013*; *LaBeau-DiMenna et al., 2012*), quantification of *tn-/-* confirmed a decrease in larval muscle diameter in both L2 (*Figure 3E,G*) and L3 individuals (*Figure 3F,G*). Thus, loss of TRIM32 compromises the glycolytic-driven growth of muscles during larval development.

As an initial assessment of glycolytic activity, we measured the extracellular acidification rate (ECAR) in *WT* or *tn-/-* muscle carcasses using the Agilent Seahorse XFe96 Analyzer. Since a major contributor to extracellular acidification is lactate excretion into medium during glycolysis (*Mookerjee and Brand, 2015*; *TeSlaa and Teitell, 2014*), we anticipated, and indeed, observed a reduction in ECAR upon loss of TRIM32 (*Figure 3H*). While quantitative, end-point assays such as ECAR, only measure a population of muscles at a single time point. For real-time measurements of glycolytic flux, the Xe96 Analyzer was used to calculate the proton efflux rate (PER). This assay specifically provides a measurement of protons derived from glycolysis by correcting for both the production of protons derived from mitochondrial reactions and for the buffering capacity of the assay medium. Briefly, the assay measures ECAR of the system under three conditions: (1) without manipulation, (2) following addition of the electron transport chain inhibitors rotenone and antimycin A, and (3) after addition of the glycolytic inhibitor 2-deoxy-D-glucose (2-DG). The end result of the assay is a measurement of acidification rate largely due to glycolytic sources. Our analysis revealed that

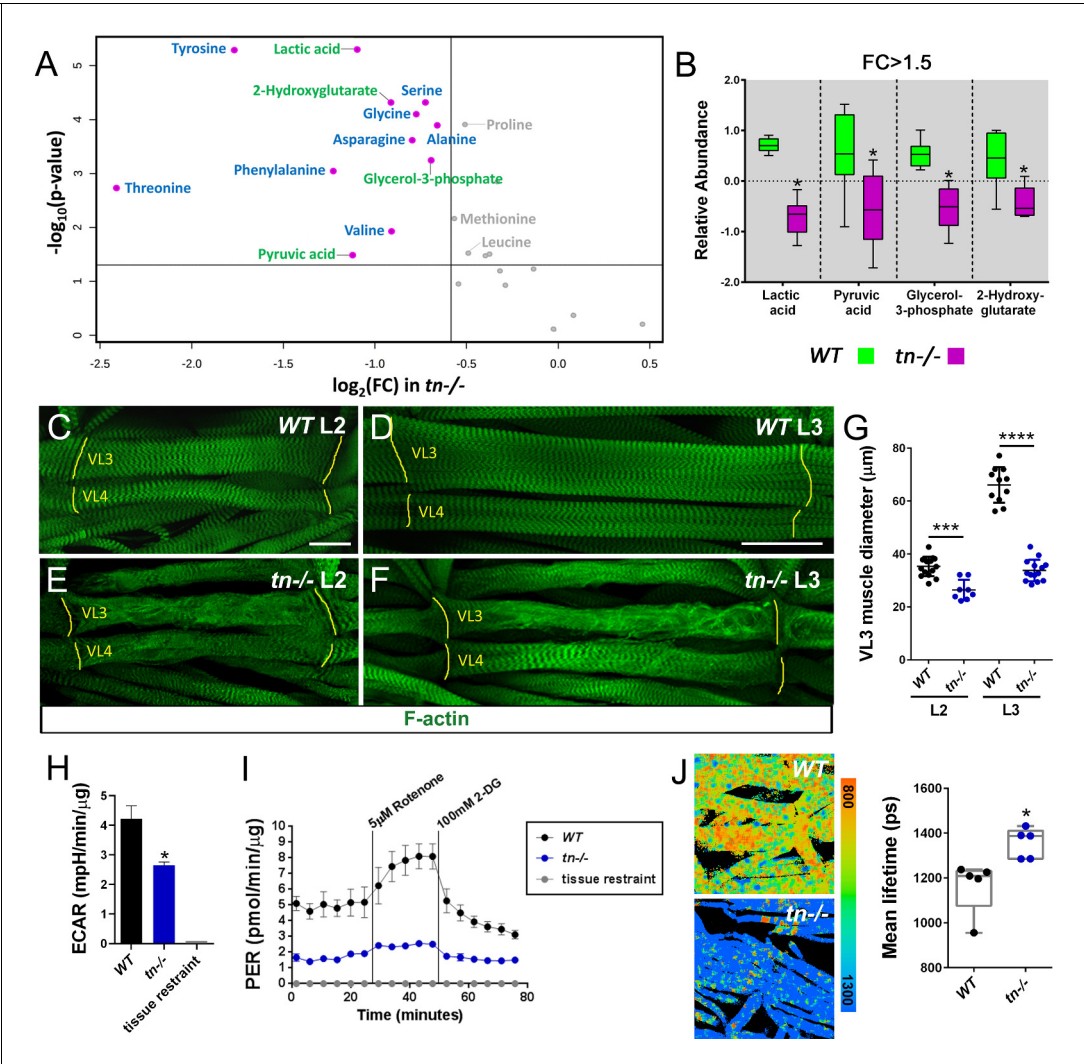

**Figure 3.** Loss of TRIM32 decreases glycolytic flux and reduces muscle tissue size. (**A**) Volcano plot illustrating fold change (FC) (log base 2) compared with p-value (- log base 10) between *WT* and *tn-/-* L3 larvae. Vertical line represents FC >1.5. Horizontal line depicts a significance level p<0.05. Metabolites that are reduced in *tn-/-* larvae include indicators of glycolytic flux (green) and amino acids (blue). Metabolites in gray are significant, but exhibit a FC <1.5. (**B**) Box and whisker plot of terminal glycolytic metabolites significantly reduced upon loss of TRIM32. N = 6. (**C–F**) Ventral longitudinal muscles 3 (VL3) and 4 (VL4) stained with phalloidin to visualize F-actin (green). (**C,D**) The stereotypical morphology of *WT* muscles is not altered as overall muscle size increases from the L2 (**C**) to the L3 (**D**) stage. (**E,F**) In addition to sarcomeric disorganization, the VL3 and VL4 muscles are noticeably smaller in *tn-/-* larvae during L2 (**E**) and L3 (**F**) development. Muscle attachment sites (MASs) are denoted by yellow lines. (**G**) Scatter plot depicting VL3 muscle diameter. The diameter of *WT* muscles increase from the L2 to the L3 stage. This cell size increase is abolished in *tn-/-*. N ≥ 8. (**H**) Bar graph shows that ECAR measurements are decreased in isolated *tn-/-* muscle carcasses compared to *WT*. N ≥ 4. (**I**) Analysis of the glycolytic rate in *WT* or *tn-/-* muscle tissue after subtraction of mitochondrial-produced acidification. This PER is diminished upon loss of TRIM32. N = 4. (**J**) NADH lifetime image comparison of *WT* and *tn-/-* muscles. Box and whisker plot shows *WT* muscles have significantly lower NADH lifetime, indicative of higher glycolytic flux, than *tn-/-*. N = 5. Mean +/- SD. (****, p<0.001; ***, p<0.01; *, p<0.05; n.s., not significant). Scale bars: 25 µm (**C,E**), 50 µm (**D,F**). The online version of this article includes the following figure supplement(s) for figure 3:

**Figure supplement 1.** Metabolite analysis upon loss of TRIM32.

glycolytic metabolism produces fewer protons in *tn-/-* muscle carcasses as compared with *WT* controls (*Figure 3I*), thus supporting our hypothesis that TRIM32 is required to support glycolytic flux.

To independently validate that loss of TRIM32 results in decreased glycolytic flux, we directly measured NADH levels within muscle tissues using fluorescence lifetime imaging (FLIM), which exploits the fluorescent characteristic of this cofactor to visualize NADH levels in cellular microenvironments (*Martin et al., 2018*; *Provenzano et al., 2009*; *Szaszák et al., 2011*; *Yaseen et al., 2017*).

Since the fluorescent lifetime of NADH (the time required for NADH to decay when exposed to 740 nm wavelength light) is longer when NADH is bound to mitochondrial enzymes than in the free state, this assay can distinguish intracellular NADH pools (*Bird et al., 2005*; *Skala et al., 2007*). This analysis revealed that *WT* muscles analyzed by FLIM showed a markedly shorter lifetime than *tn-/-* muscles (*Figure 3J*). Since free, unbound NADH is predominate in highly glycolytic cells, here we confirm a reduction in the glycolytic profile upon loss of TRIM32. These data, taken together, strongly support the hypothesis that TRIM32 maintains glycolytic flux in larval muscle tissue.

One consequence of analyzing *tn* mutant alleles is the possibility that the observed muscle defects result from loss of TRIM32 in other tissues. To rule out TRIM32-mediated systemic defects, we utilized tissue-specific RNAi approaches (*Brand and Perrimon, 1993*). First, we reconfirmed that induction of three independent RNAi constructs targeting *tn mRNA* transcripts with the *mef2*-Gal4 muscle driver produced smaller muscles (*Figure 4A,B,D*; *Brooks et al., 2016*; *Domsch et al., 2013*; *LaBeau-DiMenna et al., 2012*). Knockdown of TRIM32 in a single muscle (5053> >*tn RNAi*, asterisk) within each hemisegment was sufficient to reduce muscle cell size (*Figure 4C*). As a proxy to monitor glycolytic activity, we assayed L-lactate levels in muscle carcasses. Consistent with our metabolomics data, the relative concentration of lactate levels was decreased in *tn-/-* muscles (*Figure 4E*). Induction of *tn RNAi* in muscle tissue (*mef2 >tn RNAi*) decreased lactate levels, while this same reduction

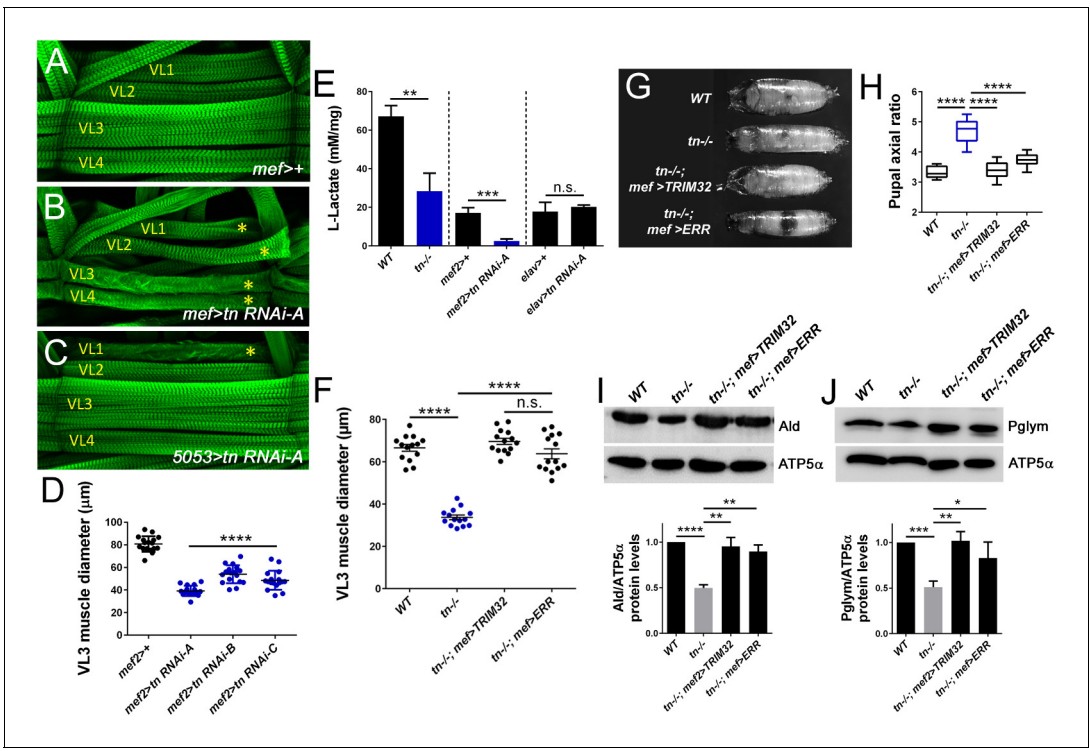

**Figure 4.** Muscle defects are cell autonomous and can be rescued upon stabilization of glycolytic enzyme levels. (**A–E**) Knockdown of TRIM32 in muscle tissue decreases muscle size and reduces lactate levels. (**A–C**) Phalloidin-labeled VL1-4 muscles in a representative hemisegment of the indicated genotypes. (**A**) *mef2>+* control muscles appear *WT*. (**B,C**) RNAi knockdown of *tn* in all muscles with *mef2*-Gal4 (**B**) or only muscle VL1 using the *5053*-Gal4 driver (**C**) show a reduction in muscle size (asterisk). (**D**) Knockdown of *tn* mRNA transcripts with three independent UAS-*tn RNAi* constructs in muscle tissue under control of the *mef2* promoter (*mef2 >tn RNAi*) show reduced VL3 muscle diameter compared to *mef2/+* VL3 muscles. N ≥ 10. (**E**) Bar graph reveals a cell autonomous role for TRIM32 in muscle tissue. L-lactate levels in muscle carcasses are decreased upon loss of TRIM32 in all tissues. Induction of *tn RNAi* in muscle, but not neuronal tissue, reduces the concentration of muscle-derived lactate. N > 8. (**F–J**) Muscle-specific expression of TRIM32 (*tn-/-, mef >TRIM32*) or ERR (*tn-/-, mef >ERR*) in a *tn-/-* background attenuates the loss of muscle size, muscle contraction, and stabilizes glycolytic protein levels. (**F**) Scatter plot shows that the reduced VL3 muscle diameter upon loss of TRIM32 is restored upon expression of TRIM32 or ERR in muscle tissue. N ≥ 10. (**G,H**) The inability to contract body wall muscles in *tn-/-* causes elongated pupae. Muscle-specific expression of TRIM32 or ERR restores muscle contraction. (**G**) Representative pupal cases of the indicated genotypes. (**H**) Quantitation of pupal axial ratios represented by a box and whisker plot. N = 10. (**I,J**) Western blots showing the relative amounts of Ald or Pglym protein relative to the ATP5α loading control. Both Ald and Pglym protein levels are stabilized upon TRIM32 or ERR expression in muscle tissue compared to *tn-/-*. N = 3. Mean +/- SD (****, p<0.001; ***, p<0.01; **, p<0.05; *, p<0.01; n.s., not significant).

did not occur upon TRIM32 knockdown in neurons (*elav >tn RNAi*) (*Figure 4E*). Here, we conclude that TRIM32-mediated regulation of glycolysis in muscle tissue is cell autonomous.

Estrogen-related receptor (ERR) is a nuclear hormone receptor that acts as a transcriptional switch in embryogenesis to induce genes required for aerobic glycolysis during larval growth (*Tennessen et al., 2011*). Therefore, we posited that genetic upregulation of carbohydrate metabolism genes via ERR may improve muscle growth and function. As a positive control, we expressed a cDNA encoding for TRIM32 in *tn-/-* muscle tissue (*tn-/-; mef >TRIM32*) and found that the muscle diameter was restored to *WT* (*Figure 4F*). This result also confirmed the cell autonomy of TRIM32. Overexpression of ERR in *tn-/-* muscles not only improved muscle size (*Figure 4F*), but also corrected the functional deficit associated with the inability of *tn-/-* to contract body wall muscles during pupal morphogenesis (*Figure 4G,H*; *Domsch et al., 2013*; *LaBeau-DiMenna et al., 2012*). Importantly, protein levels of Ald and Pglym were stabilized upon expression of ERR in *tn-/-* muscles (*Figure 4I,J*), indicating that restoration of glycolytic protein levels is sufficient to recover TRIM32-mediated growth defects.

## TRIM32 maintains amino acid pools

Our metabolomics analysis revealed that loss of TRIM32 not only disrupts the production of glycolytic intermediates, but also induces a significant decrease in eleven of the twenty amino acids (*Figure 5A*). These changes in amino acid abundance are likely due to both decreased synthesis and increased catabolism (*Figure 3—figure supplement 1D*), as we observed that *tn-/-* exhibited a>1.5 decrease in not only serine and glycine levels which are normally synthesized from glucose, but also a reduction in the anaplerotic amino acids proline and aspartic acid. Moreover, loss of TRIM32 also induced a significant depletion of alanine, which is both synthesized from and catabolized into pyruvate. Overall, the metabolomic profile of *tn-/-* indicates that disruption of glucose catabolism results in depletion of larval amino acid pools, raising the possibility that decreased amino acid availability contributes to the *tn-/-* muscle defects. We tested this possibility by supplementing the diets of both mutant and control larvae with amino acid sources. As an initial approach, we first determined if the *tn-/-* larval phenotype exhibited enhanced sensitivity to nutrient deprivation. Indeed, when reared on starvation media, the muscle diameter of *tn* mutants was smaller than muscles of control larvae raised on the same media (*Figure 5B,C,F*). In contrast, mutant larvae fed a diet consisting of only yeast extract or supplemented with all 20 amino acids exhibited no decrease in muscle diameter (*Figure 5D,E,F*). We observed similar results in the context of L3 body mass and muscle contraction during the larval to pupal transition, whereby *tn-/-* raised on an agar-only diet exhibited dramatically more severe phenotypes compared with *WT* control larvae (*Figure 5G,H,I*). Supplementation of the larval diet with yeast extract or amino acids, however, suppressed both phenotypes in *tn-/-* (*Figure 5G,H,I*). These results suggest that *tn-/-* mutants are uniquely sensitive to dietary amino acids, consistent with the smaller amino acid pool size observed upon loss of TRIM32.

## TRIM32 regulates glycolysis in a diversity of tissues

LDH expression generally correlates with LDH activity and increased glycolytic flux (*Tanner et al., 2018*; *Ždralević et al., 2017*). Only a few larval tissues show high LDH activity and exhibit elevated glycolytic rates (*Eichenlaub et al., 2018*; *Li et al., 2017*), including muscle and the larval brain (*Figure 6—figure supplement 1A*). Since we already established that *tn-/-* muscles were smaller, we wondered if growth of the larval brain also requires TRIM32. The overall size of larval brains dissected from *WT* (*Figure 6A*) or *tn-/-* (*Figure 6B*) L3 individuals showed a dramatic size difference, whereby loss of TRIM32 reduced the average area of the larval brain by ~40% (*Figure 6G*). To determine if a brain-specific reduction in TRIM32 is cell autonomous, we induced two independent *tn RNAi* lines in both neuronal (*elav >tn RNAi*) and muscle tissue (*mef2 >tn RNAi*). As expected, reduced larval brain size was observed upon RNAi knockdown of TRIM32 with *elav*-Gal4 (*Figure 6C, D,G*), but not with the *mef2* driver (*Figure 6E–G*).

Increased biomass accumulation is a primary mechanism for the hypertrophic growth of postmitotic larval muscles (*Demontis and Perrimon, 2009*), while larval brain development requires both cell growth and cell proliferation (*Hartenstein et al., 2008*). Thus, we wondered whether the TRIM32-mediated reduction in brain size was associated with altered glycolytic activity that reduced cell growth. Two experiments were performed to assess the glycolytic state in larval brain tissue.

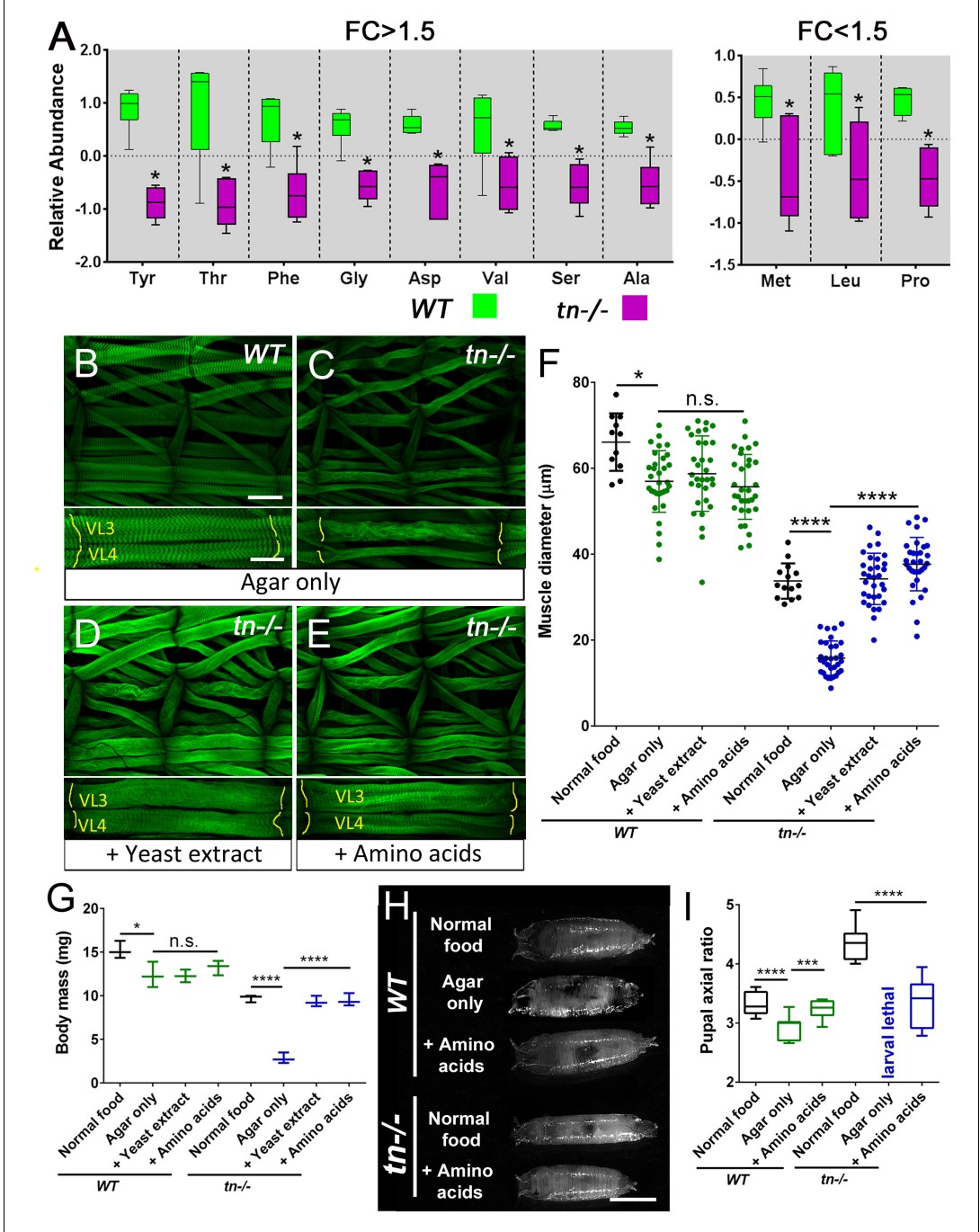

**Figure 5.** Amino acid supplementation is sufficient to improve *tn-/-* muscle mass. (A) Box and whisker plots showing the relative abundance of individual amino acids in L3 larvae with a FC >1.5 (left panel) or FC <1.5 (right panel) that are significantly reduced upon loss of TRIM32. N = 6. (B–E) Maximum intensity projections of *WT* (B) or *tn-/-* (C–E) L3 muscles stained for F-actin. Upper panel depicts two complete hemisegments and lower panel focuses on the VL3 and VL4 muscles. MASs are denoted by yellow lines. (B) An example of thinner *WT* musculature reared on agar as a sole nutritional source. (C) Muscles in larvae deficient for TRIM32 are substantially thinner when raised on agar alone. (D,E) Suppression of the reduced muscle diameter is observed in *tn-/-* muscles supplemented with total yeast extract (D) or amino acids (E) compared to *tn-/-* muscles alone. (F) Scatter plot showing the diameter of muscle VL3 in *WT* or *tn-/-* exposed to the indicated nutritional diets. N ≥ 32. (G) Average body mass measurements of *WT* or *tn-/-* L3 larvae. Ten individuals were weighed for each biological replicate that was performed in triplicate. (H) Representative pupal cases grown on the indicated diets. (I) The axial ratio (length/width) of pupal cases represented by box and whisker plots. N ≥ 10. Mean +/- SD. (****, p<0.001; *, p<0.05; n.s., not significant). Scale bars: 100 μm (A-D, upper panel), 50 μm (A-D, lower panel), 1 mm (H).

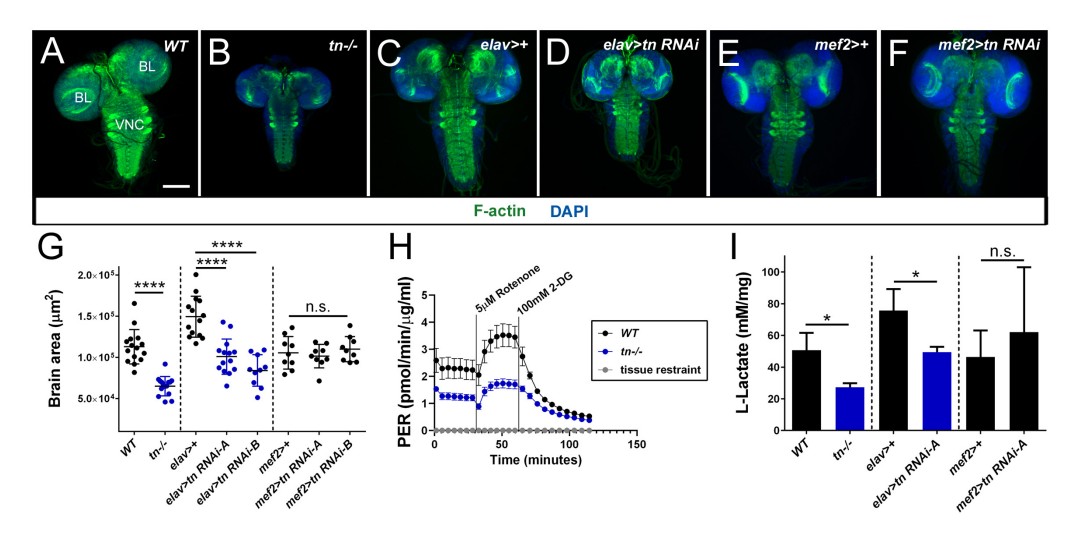

**Figure 6.** TRIM32 maintains glycolytic-mediated growth in the larval brain. (**A–F**) L3 larval brains labeled with DAPI (blue) and F-actin (green). (**A**) A representative micrograph of a *WT* larval brain showing the individual brain lobes (BL) and the ventral nerve cord (VNC). (**B**) The overall size of *tn-/-* brains is reduced due to mutations in *tn*. (**C**) Control brains expressing the pan-neuronal *elav*-Gal4 driver. (**D**) RNAi knockdown of *tn* in neurons under control of the *elav* promoter causes smaller brains. (**E,F**) Expression of *mef2*-Gal4 alone (**E**) or *mef2 >tn RNAi* in muscle tissue does not alter brain size (**F**). (**G**) Scatter plot depicting the entire brain area (including the BL and VNC) of *WT*, *tn-/-*, Gal4 driver controls, or tissue-specific *tn RNAi* knockdown brains. N ≥ 9. (**H**) Glycolytic rate assay shows a reduction in the proton efflux rate (PER) upon loss of TRIM32 in isolated L3 larval brains. The glycolytic rate is calculated after subtraction of mitochondrial-produced acidification. N = 4. (**I**) Bar graph representing L-lactate levels in isolated larval brain tissue. Only loss of TRIM32 in brain, but not muscle tissue, caused a reduction in L-lactate levels. N ≥ 15. Scale bar: 100 μm (**A–F**). Mean +/- SD. (****, p<0.001; ***, p<0.01; n.s., not significant).

The online version of this article includes the following figure supplement(s) for figure 6:

**Figure supplement 1.** Cell proliferation or cell death are not affected in *tn-/-*.

PER analysis (*Neville et al., 2018*) of individual larval brains isolated from *tn-/-* showed a reduced glycolytic rate compared to their *WT* counterparts (*Figure 6H*). Consistent with reduced substrate flux through the glycolytic pathway, L-lactate levels were also lower in dissected *tn-/-* larval brains (*Figure 6I*). Tissue-specific knockdown experiments verified that decreased lactate levels in larval brains resulted from inducing *tn RNAi* in neuronal, but not muscle tissue (*Figure 6I*). Next, we assessed how lower glycolytic activity affects overall larval brain size. Defective cell proliferation (assayed by EdU incorporation) or increased cell death (assessed using TUNEL labeling and cleaved-Caspase3 immunoreactivity) did not account for the overall brain size reduction (*Figure 6—figure supplement 1B–I*). Quantitation of individual brain cell size revealed a marked reduction upon loss of TRIM32 (*Figure 6—figure supplement 1J–M*) consistent with an increase in cell size as the primary mechanism for TRIM32-mediated tissue growth.

The elevated glycolytic rate that operates in larval muscle and brain tissue is analogous to the Warburg effect in rapidly proliferating cancer cells (*Li et al., 2017*; *Tennessen et al., 2014b*), whereby glucose metabolism is used to synthesize amino acids and other metabolites required for cell growth (*Liberti and Locasale, 2016*; *Lunt and Vander Heiden, 2011*). Thus, we hypothesized that TRIM32 could be a general regulator of highly glycolytic tumor cells. Unlike muscle or brain tissue, neither LDH activity (*Wang et al., 2016*) nor LDH-GFP expression (*Figure 7A*) were detectable in control wing discs, suggesting that this tissue does not exhibit elevated glycolytic activity. Accordingly, loss of TRIM32 did not alter the volume (*Figure 7D*) or area (*Figure 7—figure supplement 1A–C*) of mutant discs compared to *WT* control discs. Moreover, measurements of glycolytic flux using either PER assays (*Figure 7—figure supplement 1D*) or FILM analysis confirmed that TRIM32-deficient wing discs (*Figure 7F,G*) did not show altered glycolytic activity.

To examine the possibility that TRIM32 regulates the growth of highly glycolytic tumor cells, we confirmed that overexpression of the activated platelet PDGF/VEGF receptor (Pvr[act]) in *dpp*-expressing wing disc cells increased LDH-GFP expression and promoted tissue overgrowth (*Figure 7B,D*;

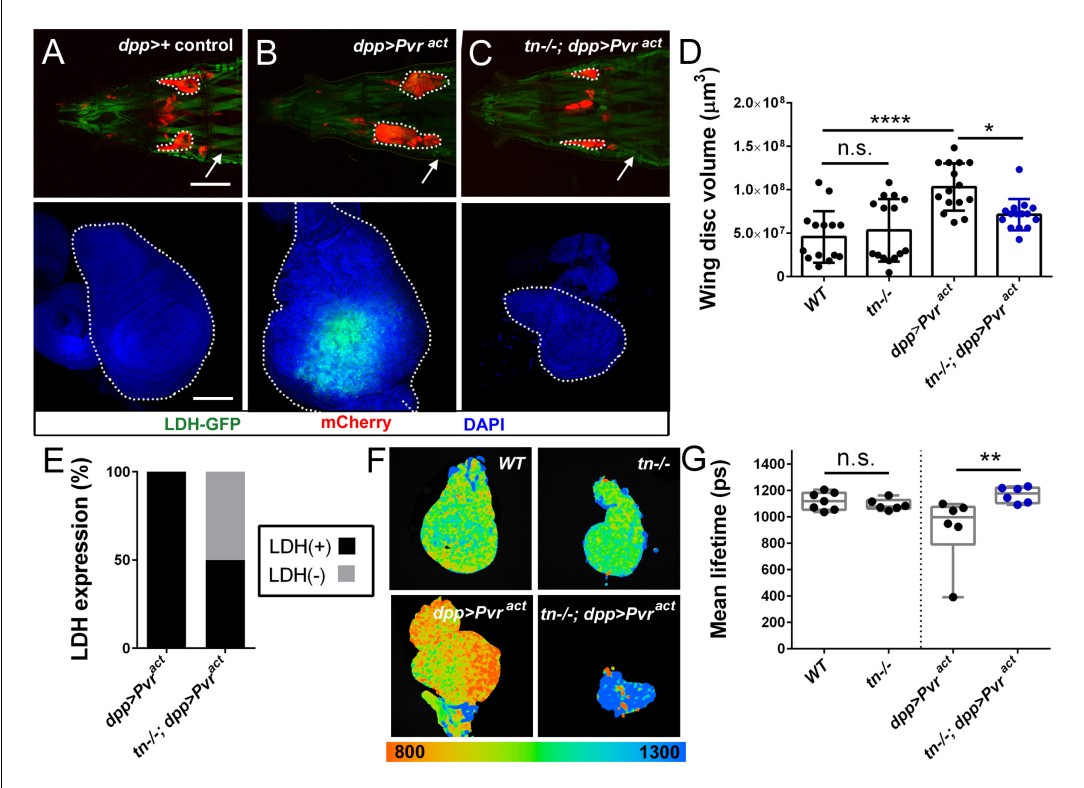

**Figure 7.** Loss of TRIM32 reduces Pvr-induced glycolytic tumor growth. (**A–C**) Either intact (upper panel; red) or isolated (lower panel, blue) wing discs from L3 larvae of the indicated genotypes. LDH-GFP is high in somatic muscles (white arrow). Wing discs are outlined (white dotted outlines). (**A**) The normal size and shape of control *dpp-Gal4/+* wing discs. (**B**) Overexpression of the activated Pvr receptor (*dpp >Pvr^act^*) causes tissue overgrowth and an increase in LDH-GFP expression (green). (**C**) Tumor growth in a *tn-/-* host is dramatically reduced in size. (**D**) Overall wing disc volumes are represented in this column plot. N ≥ 15. (**E**) Approximately 50% of LDH-GFP(+) cells induced by activated Pvr expression is reduced upon loss of TRIM32. N = 20. (**F**) Representative fluorescence lifetime micrographs of control (*WT* or *tn-/-*) or tumorous (*dpp >Pvr^act^* or *tn-/-; dpp >Pvr^act^*) wing discs. (**G**) Box and whisker plot confirms no difference in the glycolytic profile between *WT* or *tn-/-* discs. The decreased lifetime in *dpp >Pvr^act^* discs, indicative of higher glycolytic flux, is reduced upon loss of TRIM32. N = 6. Mean +/- SD. (****, p<0.001; *, p<0.05; n.s., not significant). Scale bars: 0.5 mm (A-C, upper panels), 100 μm (A-C, lower panels).

The online version of this article includes the following figure supplement(s) for figure 7:

**Figure supplement 1.** Wing disc size and glycolysis are not altered in *tn-/-*.
**Figure supplement 2.** Glucose uptake and feeding behavior upon whole animal or tissue-specific loss of TRIM32.

*Figure 7—figure supplement 1C*; *Wang et al., 2016*). Remarkably, removal of TRIM32 caused smaller tumors, effectively reducing the overall size of the wing disc (*Figure 7C,D*; *Figure 7—figure supplement 1C*). Analysis of LDH-GFP expression revealed biological variability as ~50% of TRIM32-deficient wing discs lost LDH-GFP expression (*Figure 7E*). FILM analysis validated a decrease in the glycolytic activity of tumors grown in *tn-/-* larvae compared to Pvr-induced tumors in *WT* individuals (*Figure 7F,G*). Collectively, our data show that TRIM32 is required for cell growth in both normal and tumorous glycolytic tissues, thus providing a novel molecular explanation for the upregulation of TRIM32 in multiple types of cancer cells.

## Discussion

A unique feature of *Drosophila* larval development is the inherent glycolytic nature of muscle and brain tissue (*Li et al., 2017*; *Tennessen et al., 2014b*; *Tixier et al., 2013*), which promotes biomass synthesis during this stage of rapid organismal growth. Maintenance of such a high metabolic rate predicts that enzymes are present at sufficient concentrations in the cell to mediate the rapid shunting of intermediates through the pathway (*Menard et al., 2014*). We show here that TRIM32 directly

interacts with and maintains the levels of two glycolytic enzymes. Decreased protein levels of both Ald and Pglym (and possibly other glycolytic enzymes) cripple this rapid flux, effectively blunting the generation of metabolic intermediates that contribute to anabolic synthesis necessary to sustain cell growth (*Figure 8*).

An alternative mechanism that limits cell and tissue growth is nutrient deprivation (*Ahmad et al., 2018*). In many organisms, the insulin/target of rapamycin (TOR) pathways integrate nutritional signals to physiologically control body size (*Hyun, 2013*). Implicit in this mechanism is the scaling of individual organs. Four pieces of evidence refute nutritional status as a mechanism for TRIM32-mediated tissue growth. First, the overall body size of *tn* mutant pupae is not smaller than their *WT* counterparts under starvation conditions, but instead elongated due to defective muscle contraction (*Domsch et al., 2013*; *LaBeau-DiMenna et al., 2012*; *Figure 4G,H*; *Figure 5H,I*). Second, the size of the wing disc and midgut, both tissues that show reduced growth in poorly fed larvae, is not altered upon loss of TRIM32 (*Figure 7D*; *Figure 7—figure supplement 1A–C*; *Figure 7—figure supplement 2F–H*). Surprisingly, cellular glucose uptake assayed by the non-metabolizable fluorescent glucose analog 2-[*N*-(7-nitrobenz-2-oxa-1,3-diazol-4-yl)amino]−2-deoxyglucose (2-NBDG), was normal in *tn-/-* larval brains and wing disc-derived tumors (*Figure 7—figure supplement 2A–D*), demonstrating that glucose is not a limiting substrate for glycolysis in isolated *tn-/-* tissues. Finally, even though systemic effects compromise feeding in *tn-/-* whole larvae (*Figure 7—figure supplement 2E*), food intake remained constant upon tissue-specific knockdown of TRIM32 in muscle or brains that show reduced tissue growth (*Figure 7—figure supplement 2I,J*). Collectively, these data demonstrate that TRIM32 functions in a cell autonomous manner to regulate tissue growth, independent of systemic nutritional status.

Li, et al., recently reported that pathways controlling lactate and glycerol-3-phosphate metabolism function redundantly in larval growth (*Li et al., 2019*). Removal of LDH and hence lactate production caused an increase in glycerol-3-phosphate, which was sufficient to maintain larval redox balance. Since both LDH and GPDH1 regulate redox balance necessary for maintaining high glycolytic flux to promote biomass accumulation, *Ldh*, *Gpdh1* double mutants exhibit severe growth defects with reduced brain size. Our results show that loss of TRIM32 decreases both lactate and

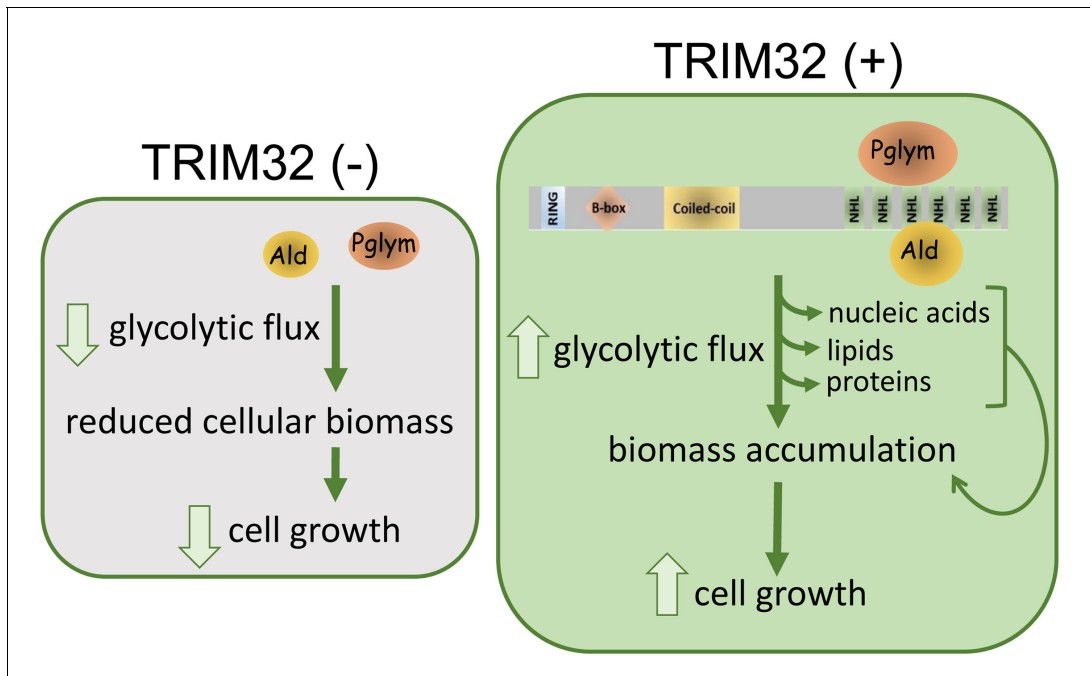

**Figure 8.** Model for TRIM32 function in the regulation of cell size. Biochemical interactions between TRIM32 and glycolytic enzymes such as Ald or Pglym cooperate in maintaining glycolytic activity for the synthesis of macromolecules required for cell growth. Loss of TRIM32 results in reduced levels of glycolytic enzymes, reduced glycolytic pathway intermediates, and compromises cell growth.

glycerol-3-phosphate levels (*Figure 3A,B*), thus mimicking the reduced carbohydrate metabolism in *Ldh, Gpdh1* double mutants.

It is not clear how mutations in a ubiquitously expressed protein such as TRIM32 result in tissue-specific diseases. One prediction is that TRIM32 differentially interacts with proteins in diverse cell types to elicit distinct biological outputs. There is strong evidence to support this hypothesis in the context of LGMD2H. TRIM32 is upregulated in proliferating satellite cells and loss of this protein prevents myotube regeneration, partially through the misregulation of NDRG and c-Myc (*Kudryashova et al., 2012*; *Mokhonova et al., 2015*; *Nicklas et al., 2012*; *Servián-Morilla et al., 2019*). Muscle-specific targets that contribute to disease progression are less clear. TRIM32-mediated deregulation of key muscle substrates, including actin, α-actinin, tropomyosin, and desmin, contribute to muscle atrophy (*Cohen et al., 2014*; *Cohen et al., 2012*; *Kudryashova et al., 2005*; *Locke et al., 2009*), but studies have not been performed to directly test this model in the context of LGMD2H. Interestingly, mammalian glycolytic type II fibers are preferentially affected over oxidative type I fibers in muscle atrophy induced by aging/starvation, as well as in Duchenne's and Becker muscular dystrophies (DMD) (*Ciciliot et al., 2013*; *Pant et al., 2015*). *TRIM32 KO* muscles also show a decrease in the glycolytic proteins GAPDH and PyK (*Mokhonova et al., 2015*), just as we observe a reduction in Ald and Pglym levels in *tn-/-* muscles, suggesting that TRIM32-mediated regulation of glycolysis may be a general mechanism that underlies some muscular dystrophies.

The multi-faceted roles exhibited by TRIM32 in muscle physiology and cancer seem quite different on the surface. However, control of glycolytic flux may be a common mode of regulation that has been overlooked. As in muscle tissue, the majority of studies on TRIM32 and cancer have focused on identifying substrates that are subject to poly-ubiquitination and subsequent proteasomal degradation. Piasy, p53 and Abi2 are known targets of TRIM32 E3 activity that regulate the proliferative balance in cancer cells (*Albor et al., 2006*; *Kano et al., 2008*; *Liu et al., 2014*). The proteolytic turnover of these proteins may affect signaling pathways independent of glycolytic TRIM32 regulation or may be a compensatory mechanism in response to metabolic shifts in which normal cells can transiently adopt cancer-like metabolism during periods of rapid proliferation.

## Limitations of this study

How does this loss of TRIM32 lead to a reduction in glycolytic enzymes? Glycolytic proteins may be substrates for TRIM32 E3 ligase activity. It seems unlikely that TRIM32 polyubiquitinates Ald or Pglym for proteasomal degradation as protein levels are not elevated upon loss of this putative E3 activity. Furthermore, co-immunoprecipitaion of higher molecular weight forms of Ald or Pglym with TRIM32 suggests a yet unidentified post-translational modification. Another possibility, which we favor, is that the NHL domain of TRIM32 serves as a scaffold for the subcellular localization of glycolytic proteins to limit diffusion of substrates during glycolysis. This does not negate, but rather expands the repertoire of TRIM32 functions.

# Materials and methods

**Key resources table**

| Reagent type (species) or resource | Designation | Source or reference | Identifiers | Additional information |
|---|---|---|---|---|
| Gene (*Drosophila melanogaster*) | *thin (tn)* | *LaBeau-DiMenna et al., 2012* | FLYB:FBgn0265356 | |
| Gene (*D. melanogaster*) | *Aldolase 1 (Ald)* | | FLYB:FBgn0000064 | |
| Gene (*D. melanogaster*) | *Phosphogylcerate mutase 78 (Pglym)* | | FLYB:FBgn0014869 | |
| Genetic reagent (*D. melanogaster*) | *w¹¹¹⁸* | Bloomington *Drosophila* Stock Center (BDSC) | BL3605 | |
| Genetic reagent (*D. melanogaster*) | *tnΔA* | *LaBeau-DiMenna et al., 2012* | | |

*Continued on next page*

*Continued*

| Reagent type (species) or resource | Designation | Source or reference | Identifiers | Additional information |
|---|---|---|---|---|
| Genetic reagent (*D. melanogaster*) | *CyO, Tb/Sco* | BDSC | BL36335 | |
| Genetic reagent (*D. melanogaster*) | *mef*-Gal4 | BDSC | BL27390 | |
| Genetic reagent (*D. melanogaster*) | *elav*-Gal4 | BDSC | BL458 | |
| Genetic reagent (*D. melanogaster*) | *5053*Gal4 | BDSC | BL2702 | |
| Genetic reagent (*D. melanogaster*) | *UAS-tn RNAi-A* | Vienna *Drosophila* Resource Center (VDRC) | v19290 | |
| Genetic reagent (*D. melanogaster*) | *UAS-tn RNAi-B* | BDSC | BL31588 | |
| Genetic reagent (*D. melanogaster*) | *UAS-tn RNAi-C* | VDRC | v19291 | |
| Genetic reagent (*D. melanogaster*) | *dpp-UAS-mcherry, LDH-GFP* | **Wang et al., 2016** | | |
| Genetic reagent (*D. melanogaster*) | UAS-Pvr$^{act}$ | **Wang et al., 2016** | | |
| Genetic reagent (*D. melanogaster*) | *LDH-optGFP* | Materials and methods | | |
| Genetic reagent (*D. melanogaster*) | UAS-*TRIM32* | **LaBeau-DiMenna et al., 2012** | | |
| Genetic reagent (*D. melanogaster*) | UAS-*ERR*-FLAG | Materials and methods | | |
| Antibody | anti-TRIM32 (guinea pig polyclonal) | **LaBeau-DiMenna et al., 2012** | | (1:500) |
| Antibody | anti-Pglym (rabbit polyclonal) | **Sullivan, 2003** | | (1:1000) from Jim Vigoreaux |
| Antibody | anti-Ald (rabbit polyclonal) | **Sullivan, 2003** | | (1:1000) from Jim Vigoreaux |
| Antibody | anti-Tm (rat monoclonal) | Babraham Institute | MAC141 | (1:500) |
| Antibody | anti-hALD | Biorad | VPA00226 | (1:1000) |
| Antibody | anti-hPGAL1 | Cell Signaling | D3J9T | (1:1000) |
| Antibody | anti-ATP5α (mouse monoclonal) | Abcam | Catalog# ab14748 | (1:10000) |
| Antibody | anti-Cleaved Caspase-3 | Cell Signaling | Catalog# 9661 | (1:100) |
| Antibody | Alexa 488 secondaries | Thermo Fisher | Catalog# A12379 | (1:400) |
| Antibody | Rabbit IgG HRP Linked Whole Ab | GE Healthcare | NA934-1ML | (1:3000-1:5000) |
| Antibody | Mouse IgG HRP Linked Whole Ab | GE Healthcare | NA931-1ML | (1:3000-1:5000) |
| Recombinant DNA reagent | pGEX-5X-2_TRIM32_NHL | Materials and methods | | nucleotides 3231–4062 |
| Recombinant DNA reagent | pT7HMT_Ald | Materials and methods | | His-tagged Ald |
| Recombinant DNA reagent | pT7HMT_Pglym | Materials and methods | | His-tagged Pglym |
| Recombinant DNA reagent | pT7HMT_SCIN | **Ricklin et al., 2009** | | His-tagged SCIN |

*Continued on next page*

*Continued*

| Reagent type (species) or resource | Designation | Source or reference | Identifiers | Additional information |
|---|---|---|---|---|
| Sequence-based reagent | pGEX-5X-2_NHL_5'F | Integrated DNA Technologies (IDT) | | Oligonucleotide GGGATCCCCGGAATTCCCCTGC GCAAGCGCCAGCAGCTGTTC |
| Sequence-based reagent | pGEX-5X-2_NHL_5'R | Integrated DNA Technologies (IDT) | | Oligonucleotide ATAAGAATGCGGCCGCCT GGCGCTTGCGCAGGTACACCTG |
| Sequence-based reagent | pT7HMT_Tm2_5'F | Integrated DNA Technologies (IDT) | | Oligonucleotide ACAGGATCCATGG ACGCCATCAAGAAGAAG |
| Sequence-based reagent | pT7HMT_Tm2_5'R | Integrated DNA Technologies (IDT) | | Oligonucleotide AAGGAAAAAAGCGGCCG CTTAGTAGCCAGCCAATTCGGC |
| Sequence-based reagent | pT7HMT_Ald_5'F | Integrated DNA Technologies (IDT) | | Oligonucleotide ACAGGATCCATGACGA CCTACTTCAACTACC |
| Sequence-based reagent | pT7HMT_Ald_5'R | Integrated DNA Technologies (IDT) | | Oligonucleotide AAGGAAAAAAGCGGCCGC TCAATACCTGTGGTCATCCAC |
| Sequence-based reagent | pT7HMT_Pglym_5'F | Integrated DNA Technologies (IDT) | | Oligonucleotide CAGGGGTCGACAATGG GCGGCAAGTACAAGATC |
| Sequence-based reagent | pT7HMT_Pglym_5'R | Integrated DNA Technologies (IDT) | | Oligonucleotide AAGGAAAAAAGCGGCCGC TTACTTGGCCTTGCCCTGGGC |
| Sequence-based reagent | UAS—2xFLAG-ERR_5'F | Integrated DNA Technologies (IDT) | | Oligonucleotide AGCGGCCGCCATGGACTACAA GGACGACGATGACAAGGGTGACT ACAAGGACGACGATGACAAGG GTATGTCCGACGGCGTCAGCATC |
| Sequence-based reagent | UAS—2xFLAG-ERR_3'F | Integrated DNA Technologies (IDT) | | Oligonucleotide AGCGGCCGCTTATCAC CTGGCCAGCGGCTCGAGC |
| Commercial assay or kit | ATP Determination Kit | Molecular Probes | Catalog# A22066 | |
| Commercial assay or kit | Bradford Assay kit | Bio-Rad | Catalog# 5000001 | |
| Commercial assay or kit | RNAeasy Mini Kit (50) | Qiagen | Catalog# 74104 | |
| Commercial assay or kit | DeadEnd Fluorometric TUNEL System | Promega | Catalog# G3250 | |
| Commercial assay or kit | Click-iT EdU Cell Proliferation Kit for Imaging, Alexa Fluor 488 dye | Thermo Fisher | Catalog# C10337 | |
| Commercial assay or kit | ECL Plus Western Blotting Detection kit | Thermo Fisher | Catalog# 32132 | |
| Commercial assay or kit | SuperScript VILO cDNA Synthesis Kit | Invitrogen | Catalog# 11754050 | |
| Commercial assay or kit | EnzyChromTM L-Lactate Assay Kit | BioAssay Systems | Catalog# ECLC-100 | |
| Commercial assay or kit | Power UP SYBR Green Master mix | Applied Biosystems | Catalog# A25741 | |
| Chemical compound, drug | DAPI (4',6-diamidino-2-phenylindole, Dihydrochloride) | Thermo Fisher | Catalog# D1306 | |

*Continued on next page*

*Continued*

| Reagent type (species) or resource | Designation | Source or reference | Identifiers | Additional information |
|---|---|---|---|---|
| Chemical compound, drug | 2-NBDG | Cayman Chemicals | Catalog# 186689-07-6 | |
| Chemical compound, drug | Erioglaucine disodium salt | Milipore Sigma | Catalog# 861146 | |
| Chemical compound, drug | Formaldehyde, 16% Methanol-free, ultra-pure EM Grade | Polyscience | Catalog# 1881lawr4 | |
| Chemical compound, drug | Triton X-100 | Sigma-Aldrich | Catalog# 9002-93-1 | |
| Chemical compound, drug | Tween20 | Sigma-Aldrich | Catalog# 9005-64-5 | |
| Chemical compound, drug | Glycerol | Fisher | Catalog# BP229-1 | |
| Chemical compound, drug | Methanol | Fisher | Catalog# A412P-4 | |
| Chemical compound, drug | Bromophenol-blue | Amresco | Catalog# 0449–25G | |
| Chemical compound, drug | DTT (1,4-Dithiothreitol) | Sigma-Aldrich | Catalog# 3483-12-3 | |
| Chemical compound, drug | Tris base | Fisher | Catalog# BP152-5 | |
| Chemical compound, drug | Sodium Chloride | Fisher | Catalog# BP358-212 | |
| Chemical compound, drug | Hydrochloric acid | Fisher | Catalog# A144-50/A144S212 | |
| Chemical compound, drug | Potassium Chloride | Fisher | Catalog# BP366-500 | |
| Chemical compound, drug | Magnesium Chloride | Fisher | Catalog# M-33 | |
| Chemical compound, drug | Sodium Bicarbonate | Fisher | Catalog# S233-500 | |
| Chemical compound, drug | Calcium Chloride Dihydrate | Fisher | Catalog# C-79 | |
| Chemical compound, drug | Sodium Dodecyl Sulphate | Fisher | Catalog# BP166-500 | |
| Chemical compound, drug | Sucrose | Fisher | Catalog# S3-500 | |
| Chemical compound, drug | Agar, Powder/Flakes | Fisher Scientific | Catalog# BP1423-500 | |
| Chemical compound, drug | L-amino acids | Sigma-Aldrich | Catalog# 200-157-7 | |
| Chemical compound, drug | Yeast Extract Hy-Yest 412 | Kind gift from Dr. Lawrence Davis | N/A | |
| Chemical compound, drug | HEPES | Fisher | Catalog# BP310-100 | |
| Chemical compound, drug | TEMED | Santa Cruz | Catalog# SC-29111 | |
| Chemical compound, drug | Ammonium Persulfate | Fisher | Catalog# BP179-100 | |
| Chemical compound, drug | HisPur Ni-NTA Magnetic Beads | Thermo Fisher | Catalog# 88831 | |

*Continued on next page*

*Continued*

| Reagent type (species) or resource | Designation | Source or reference | Identifiers | Additional information |
|---|---|---|---|---|
| Chemical compound, drug | Cyanogen bromide activated Sepharose 4B | Sigma-Aldrich | Catalog# 68987-32-6 | |
| Software, algorithm | Graphpad Prism 7.00 | GraphPad Software | https://www.graphpad.com/ | |
| Software, algorithm | ImageJ | NIH | https://imagej.nih.gov/ij/ | |
| Software, algorithm | Adobe Photoshop | Adobe | N/A | |
| Software, algorithm | Zen black | Zeiss | N/A | |
| Other | Zeiss 700 confocal microscope | Zeiss | N/A | |

## Fly genetics

Fly stocks were reared on standard cornmeal media at 25°C unless otherwise indicated. The *WT* control strain was *w^1118* [Bloomington (BL) *Drosophila* Stock Center; BL3605]. The *tnΔA* mutation (*LaBeau-DiMenna et al., 2012*) was maintained over the *CyO, Tb/Sco* (BL36335) balancer chromosome. Non-Tb individuals from the *tnΔA/CyO, Tb^1* stock were used for *tn-/-* analysis. The following Gal4 and/or RNAi lines were used: *mef2*-Gal4 (BL27390), *elav*-Gal4 (BL458), *5053*-Gal4 (BL2702), UAS-*tn* RNAi-A [Vienna *Drosophila* Resource Center (VDRC); v19290], UAS-*tn* RNAi-B (BL31588), and UAS-*tn* RNAi-C (v19291). UAS-*TRIM32* was previously described (*LaBeau-DiMenna et al., 2012*). The *UAS-2XFLAG-dERR* transgene was generated by amplifying the *dERR* cDNA using the oligos listed in *Table 1*. The resulting PCR product was sequenced, inserted into the NotI site of pUAST-attP, and injected to a strain containing the attP40 site by BestGene Inc (Chino Hills, CA). The transgenic *dpp-Gal4, UAS-mCherry, LDH-GFP* and *UAS-Pvr^act* flies were a kind gift from U. Banerjee (*Wang et al., 2016*). The *LDH-GFP* line (*Figure 6—figure supplement 1A*) was generated using a previously described *pLdh* genomic rescue construct (*Li et al., 2017*). Briefly, GFP was inserted at the 3′ end of the *Ldh* coding region using a PCR based method. The plasmid was injected into the strain attP40w by Rainbow Transgenics (Camarillo, CA) and the F1 generation was screened for transgene integration at the attP40 docking site. This transgene is capable of rescuing the *Ldh* mutant larval lethal phenotype (Chawla and Tennessen, in preparation).

## Immunostaining and microscopy

*Muscle.* Synchronized L2 or wandering L3 larvae were rinsed with 0.7% (w/v) NaCl/0.1% Triton, filleted in ice cold 1X PBS on Sylgard plates, and fixed in 4% (v/v) formaldehyde (Fisher) followed by three washes with 0.5% PBT. Phalloidin 488 or 594 was used to label F-actin (1:400, Molecular Probes). *Other tissues.* Wing discs, larval brains, or midguts were isolated from wandering L3 larvae and fixed for 25 min in 4% formaldehyde in PBS. Both discs and brains were stained overnight with DAPI (1:400) at 4°C. Midguts were stained with Phalloidin 488 (1:400, Molecular Probes). Anti-Caspase-3 (1:100, Cell Signaling Technology, Danvers, MA) staining on brains was performed overnight at 4°C followed by labeling with Alexa Fluor anti-rabbit 568 secondary antibody (1:400). Either muscle fillets, isolated brains or wing discs were mounted in anti-fade mounting medium (190% glycerol, 0.5% n-propyl gallate in 20 mM Tris buffer, pH = 8.0) and imaged using a Zeiss 700 confocal microscope.

## Molecular biology

The NHL region of *Drosophila* TRIM32 (nucleotides 3231–4062) was PCR amplified using Phusion polymerase (ThermoFisher Scientific), digested with EcoRI and NotI, and ligated into the pGEX-5X-2 expression vector containing an N-terminal GST tag. Tm2, Ald, and Pglym were amplified by PCR and subcloned into the pT7HMT protein expression vector with either SalI/NotI or BamHI/NotI. Primers used for PCR amplification are included in *Table 1*.

**Table 1.** Statistics Summary.

| Panel | Graph type | N value | Statistical test used | Precision | p-value |
|---|---|---|---|---|---|
| *Figure 2E,F* | Bar graphs | Pool of 5 larvae per genotype (N = 3 biological replicates and N = 3 technical replicates) | Unpaired t-test | Mean +/- SD | p<0.05 |
| *Figure 3A* | Volcano plot | Pool of 25 larvae per genotype (N = 6 biological replicates) | Univariate fold change and t-test analysis | N/A | FC > 1.5 p<0.05 |
| *Figure 3B* | Box and whisker plot | Pool of 25 larvae per genotype (N = 6 biological replicates) | Unpaired t-test | Min to max | p<0.05 |
| *Figure 3G* | Scatter plot | N ≥ 8 | One-Way ANOVA Kruskal-Wallis test | Mean +/- SD | p<0.001, p<0.01 |
| *Figure 3H* | Bar graph | N = 4 | Unpaired t-test | Mean +/- SD | p<0.05 |
| *Figure 3I* | Mean and error plot | N = 4 | Holm-Sidak t-test | Mean +/- SEM | p<0.05 |
| *Figure 3J* | Box and whisker plot | N = 5 | Unpaired t-test | Min to max | p<0.05 |
| *Figure 4D* | Scatter plot | N ≥ 10 | One-Way ANOVA Kruskal-Wallis test | Mean +/- SD | p<0.001 |
| *Figure 4E* | Bar graph | Pool of at least eight larvae per genotype (N = 3 biological replicates and N = 3 technical replicates) | One-Way ANOVA Kruskal-Wallis test | Mean +/- SD | p<0.005, p<0.01, n.s. |
| *Figure 4F* | Scatter plot | N ≥ 10 | One-Way ANOVA Kruskal-Wallis test | Mean +/- SD | p<0.001 |
| *Figure 4H* | Box and whisker plot | N = 10 | One-Way ANOVA Kruskal-Wallis test | Mean +/- SD | p<0.001 |
| *Figure 4I,J* | Bar graph | Pool of 5 larvae per genotype (N = 3 biological replicates and N = 3 technical replicates) | One-Way ANOVA Kruskal-Wallis test | Mean +/- SD | p<0.001, p<0.005, p<0.05, p<0.01 |
| *Figure 5A* | Box and whisker plot | Pools of 25 larvae (N = 6 biological replicates) | Unpaired t-test | Min to max | p<0.05 |
| *Figure 5F* | Scatter plot | N ≥ 32 | One-Way ANOVA Kruskal-Wallis test | Mean +/- SD | p<0.05, p<0.001, n.s. |
| *Figure 5G* | Column graph | Pool of 10 larvae per genotype subjected to each condition (N = 3) | One-Way ANOVA Kruskal-Wallis test | Mean +/- SD | p<0.05, p<0.001, n.s. |
| *Figure 5I* | Box and whisker plot | N ≥ 10 | One-Way ANOVA Kruskal-Wallis test | Min to max | p<0.001 |
| *Figure 6G* | Scatter plot | N ≥ 9 | One-Way ANOVA Kruskal-Wallis test | Mean +/- SD | p<0.001, n.s. |
| *Figure 6H* | Mean and error plot | N = 4 | Holm-Sidak t-test | Mean +/- SEM | p<0.05 |
| *Figure 6I* | Bar graph | Pool of at least 15 larvae per genotype (N = 3 biological replicates and N = 3 technical replicates) | One-Way ANOVA Kruskal-Wallis test | Mean +/- SD | p<0.005, p<0.01, n.s. |
| *Figure 7D* | Scatter plot | N = 15 | One-Way ANOVA Kruskal-Wallis test | Mean +/- SD | p<0.001, p<0.05, n.s. |
| *Figure 7E* | Bar graph | N = 20 | Unpaired t-test | Mean | N/A |
| *Figure 7G* | Box and whisker plot | N = 6 | One-Way ANOVA Kruskal-Wallis test | Min to max | p<0.05, n.s. |
| *Figure 2—figure supplement 1C* | Bar graph | Pool of 5 larvae per genotype (N = 3 biological replicates and N = 3 technical replicates) | Unpaired t-test | Mean +/- SD | p<0.05 |
| *Figure 2—figure supplement 1F,G* | Bar graph | Pool of 5 larvae per genotype (N = 3 biological replicates and N = 3 technical replicates) | Unpaired t-test | Mean +/- SD | n.s. |

*Table 1 continued on next page*

*Table 1 continued*

| Panel | Graph type | N value | Statistical test used | Precision | p-value |
|---|---|---|---|---|---|
| *Figure 3—figure supplement 1B* | Box and whisker plot | N = 6 chambers with five larvae per genotype | Unpaired t-test | Min to max | p<0.05 |
| *Figure 3—figure supplement 1C* | Bar graph | Pool of 25 larvae per genotype (N = 3 biological replicates and N = 3 technical replicates) | Unpaired t-test | Mean +/- SD | p<0.001 |
| *Figure 6—figure supplement 1D* | Scatter plot | N ≥ 9 | Unpaired t-test | Mean +/- SD | n.s. |
| *Figure 6—figure supplement 1M* | Scatter plot | N > 300 cells measured in 10 brains of each genotype | Unpaired t-test | Mean +/- SD | p<0.001 |
| *Figure 7—figure supplement 1C* | Scatter plot | N ≥ 20 | One-Way ANOVA Kruskal-Wallis test | Mean +/- SD | p<0.001, n.s. |
| *Figure 7—figure supplement 1D* | Mean and error plot | N = 10 | Holm-Sidak t-test | Mean +/- SEM | n.s. |
| *Figure 7—figure supplement 2E* | Bar graph | N = 9 | Unpaired t-test | Mean +/- SD | p<0.005 |
| *Figure 7—figure supplement 2H* | Scatter plot | N = 8 | Unpaired t-test | Mean +/- SD | n.s. |
| *Figure 7—figure supplement 2J* | Bar graph | Pool of at least 15 larvae per genotype (N = 3 biological replicates and N = 3 technical replicates) | One-Way ANOVA Kruskal-Wallis test | Mean +/- SD | p<0.05, n.s. |

## Protein expression and purification

Protein expression was performed in *E. coli* BL21 cells. A single colony was inoculated into 100 mL of LB media supplemented with the appropriate antibiotic at 37°C, incubated overnight and then diluted into 1L of Terrific Broth. Protein expression was induced at $OD_{600} = 0.6$ after the addition of Isopropyl β-D-1-thiogalactopyranoside (IPTG) to a final concentration of 1 mM (*Geisbrecht et al., 2006*). After 18 hr of incubation at 18°C, cells were centrifuged and were lysed using a microfluidizer for cell lysis. After centrifugation, the GST_TRIM32_NHL tagged protein was purified using GST affinity chromatography (GE Healthcare), further purified using size exclusion and ion-exchange Chromatography, and concentrated using 30K Amicon centrifuge columns (Millipore). The large scale expression and purification of Tm2, Ald, and Pglym was similar but used $Ni^{2+}$ column for the initial purification step for His-tag binding. The purified proteins were stored at −80°C until use.

## Crystallization, X-ray diffraction, structure solution, and refinement

Crystals of a recombinant form of the NHL-repeat region of *Drosophila* TRIM32 were obtained by vapor diffusion of hanging drops. Briefly, a sample of purified protein was buffer exchanged into double-deionized water and concentrated to 5 mg/ml. Crystals were obtained from 2 µl droplets that had been established by mixing 1 µl of protein with 1 µl of precipitant solution [0.1M HEPES (pH 7.8), 0.2 M NaCl, and 25% (v/v) PEG-3350] and incubating over 500 µl of precipitant solution at 20°C. Rod-shaped crystals appeared within 2–3 days and grew to their full size within 1–2 weeks. Single crystals were harvested and briefly soaked in a cryopreservation buffer consisting of precipitation solution supplemented with an additional 20% (v/v) glycerol. Monochromatic X-ray diffraction data were collected at beamline 22-BM of the Advanced Photon Source at Argonne National Laboratory using incident radiation of λ = 1.000 Å. Reflections were indexed, integrated, and scaled using the HKL2000 software suite (*Otwinowski and Minor, 1997*). Initial phases were obtained by molecular replacement using program PHASER (*McCoy et al., 2007*) and a poly-alanine model derived from chain A of PDB entry 1Q7F. The model was constructed and refined through an iterative process consisting of automated building and refinement in PHENIX (*Adams et al., 2002*; *Zwart et al., 2008*), coupled with manual inspection and modifications. The final model consists of 292 protein residues, 85 ordered solvent molecules, and six ligand molecules. 94% of the protein residues occupy favored regions in the Ramachandran plot, with an additional 4% in allowed regions.

Additional information on data collection and model statistics may be found in *Figure 1—source data 1*. The final model and structure factors have been deposited in the PDB under accession code 6D69.

## Mass spectrometry (MS) analysis

L3 larvae were homogenized in lysis buffer [50 mM Tris-HCl (pH 7.5), 100 mM NaCl, 10% (v/v) glycerol and 1 mM EDTA] plus inhibitors [1 mM Na3VO4, 5 mM NPPS, 2 mM PMSF, 2 ug/ml Leupeptin, 10 μM MG132, 1x Halt Pro inhibitor cocktail (Roche)]. 10 μg of purified TRIM32_NHL protein was coupled to Cyanogen bromide-activated-Sepharose 4B beads (Sigma-Aldrich), incubated for 2 hr at 4°C with 100 mg of lysates prepared from L3 larvae. Bead-protein complexes were washed 3X with wash buffer (50 mM Tris-HCl (pH 7.5), 150 mM NaCl, 1 mM EDTA and 1% (v/v) Triton]. Control pull-downs were performed using beads alone. Beads containing protein complexes were sent to Oklahoma State University for MS analysis. Statistical analysis was performed using Perseus MaxQuant (*Cox and Mann, 2012*; *Tyanova et al., 2016*).

## In vitro binding assay

To assess the interaction with TRIM32_NHL, 10 μg of purified candidate proteins (Tm2, Ald or Pglym) or the negative control (SCIN) were immobilized on $Ni^{2+}$-NTA magnetic beads (ThermoFisher Scientific) for 1.5 hr followed by incubation with 10 μg of dTRIM32 NHL for 30 min at 4°C on a rotating platform. The complexes were washed 6x [300 mM NaCl, PBS (pH = 7.0) + 1% (v/v) Triton]. The bound proteins were eluted by boiling at 100°C in 6X Laemmli buffer for 10 min. The binding was analyzed on 12% sodium dodecyl sulfate polyacrylamide gel electrophoresis (SDS-PAGE) under reducing conditions followed by Coomassie Blue Staining.

## Co-Immunoprecipitation

For lysate preparation, *WT* third-instar larvae were homogenized in lysis buffer (50 mM Tris-HCl pH 7.5, 100 mM NaCl, 10% glycerol, 1% Triton, 1 mM EDTA) containing 1X Halt Protease Inhibitor (Thermo fisher Scientific) and 10 μM MG132. The lysate was precleared via centrifugation at 13,000 x for 20 min at 4°C and the resulting protein concentration was measured using the Bradford Assay (Biorad, Hercules, CA). 500 μg of the lysate was set aside for input analysis. For immunoprecipitations, 20 μL of Protein A Sepharose 4b beads (Thermo Fisher Scientific) were washed twice with the lysis buffer. The beads were then incubated with 15 μg of *Drosophila* anti-TRIM32 antibody overnight at 4°C. The antibody conjugated beads were then mixed with 2 mg of lysate and incubated for 2 hr at 4°C on a rotating wheel. Post incubation, the resin was centrifuged at 1000 x g for 3 min washed five times with wash buffer (1M NaCl, 1%Triton, PBS). 30 μL of 6X Laemmli buffer was used to elute bound proteins off the resin and these samples were denatured by heating at 100°C for 10 min. Eluted protein samples were subjected to 10% SDS PAGE gel electrophoresis and transferred to polyvinyl difluoride (PVDF) membranes (Pierce Biotechnology, Inc, Waltham, MA) for Western blotting with the following primary antibodies: rabbit anti-*Drosophila* Ald [1:1000, (*Sullivan, 2003*)], rabbit anti- *Drosophila* Pglym [1:1000, (*Sullivan, 2003*)], guinea pig anti-*Drosophila* TRIM32 [1:500, (*LaBeau-DiMenna et al., 2012*)], rabbit anti-human ALD [1:1000, Biorad, Hercules, CA], rabbit anti-human PGAM1 [1:1000, Cell Signaling, Danvers, MA]. Post-treatment with the primary antibodies, blots were washed thoroughly with the wash buffer (TBS+ 0.1%Tween) and incubated with HRP conjugated and fluorescent secondary antibodies (dilution 1:5000) for 2 hr at room temperature. Protein detection was performed using the Prometheus ProSignal Pico western blotting detection kit (Genesee Scientific).

## Metabolomics analysis

Pooled *WT* or *tn-/-* L3 larvae (25 each) were selected and washed with NaCl/0.1% Triton to remove food or debris. Each batch of pooled larvae were flash frozen in liquid nitrogen and sent to the University of Utah Metabolomics Core. Metabolites were extracted and derivatized before gas chromatography-mass spectrometry (GC-MS) analysis with an Agilent 5977B GC-MS. Data was collected using MassHunter software and metabolite identity was determined using MassHunter Quant. The metabolite data was normalized to standards, parsed, and Metaboanalyst 3.0 was used for statistical analysis and data processing. Independently, groups of 10 larvae were weighed on an analytical

scale to determine the difference in body mass between *WT* and mutant cohorts before analysis. Six biological replicates were performed for each genotype.

## In vivo feeding assays

To assess the impact of amino acid supplements on muscle mass in *WT* and *tn-/-*, embryos were reared from hatching until analysis at the L3 stage at 25˚C on three different food conditions in 60 mm petri dishes: (1) Agar only control: 2.25% (w/v) agar in distilled water; (2) Yeast extract powder (mixture of amino acids, carbohydrates, peptides and soluble vitamins): 15% (w/v) yeast extract powder added to 2.25% (w/v) agar; or (3) Amino acids: 20 individual amino acids were individually weighed and added to 2.25% (w/v) agar (*Lee and Micchelli, 2013*). Both *WT* and *tn-/-* L3 larvae were dissected, stained with phalloidin, and analyzed using confocal microscopy for each condition. Note that agar contains two polysaccharides, agarose and agaropectin. Thus, the larvae are starved, but not completely devoid of nutrients.

## ATP assay

25 L3 larvae were pooled and homogenized in 100 µl of extraction buffer [6 M guanidine HCL, 100 mM Tris (pH 7.8), 4 mM EDTA], boiled for 5 min, and centrifuged at 13,000 x g for 5 min at 4˚C. Protein concentrations were measured using the Bradford Assay kit (Biorad, Hercules, CA). ATP levels were determined using an ATP Determination Kit (Molecular Probes) as described in *Tennessen et al. (2014a)*. 100 µl assays were performed in a 96 well plate and the luminescence was measured using Perkin Elmer EnSpire Multimode plate reader. Each sample was processed in triplicate and read in triplicate. The amount of ATP was normalized to total protein concentration.

## Quantitation and statistical analysis

*Muscle diameter measurements.* Z-stack images for each fillet were converted to a maximum intensity projection. The Polylines plugin from ImageJ was used to measure the muscle width of all VL3 muscles from dissected L2 and L3 individuals. *Pupal case axial ratio determination.* Pupa of the appropriate genotype were removed from vials, oriented dorsal side up, and attached to slides using a small drop of nail polish. Images were taken with a Leica M165 FC Stereomicroscope. Length and width measurements for each pupae were performed in ImageJ using the line and measure functions. Values were put into an Excel spreadsheet and the axial ratio (length/width) was calculated for each individual. The raw data was imported into Graphpad Prism 6.0 and graphed as a box and whiskers plot. $N \geq 10$ for each genotype. *Brain and midgut analysis.* Z-stack images of isolated brain or midgut tissue areas were measured in ImageJ using the outline and measure functions $N \geq 8$ for each genotype. Brain lobe cell size was determined using the analyze particle function after thresholding in ImageJ. $N > 300$ cells in 10 individual brain lobes for each genotype. *Larval mass measurements.* For each genotype and condition tested, pools of 10 larvae weighed on a digital scale. The average of each pool (N = 3) was plotted. *Wing disc-associated tumor analysis.* Z-stack images of each wing disc were used to measure the area and volume. For area measurements, the single plane that contained the maximum area for each disc was fully outlined using the free draw tool followed by the measure command in ImageJ. For volume quantitation, each disc in each Z-section was outlined, subjected to thresholding, and the stack command was used to measure the stacked volume. *Statistics.* Raw data was imported to GraphPad Prism 6.0 for statistical analysis and graph generation. All error bars represent mean ± standard deviation (SD). Statistical significances were determined using either student t-tests, Mann-Whitney tests or one-way ANOVA. Differences were considered significant if $p < 0.05$ and are indicated in each figure legend.

## Western blots

Five whole L3 larvae were homogenized in 3X SDS sample buffer [188 mM Tris-HCl (pH 6.8), 3% (w/v) SDS, 30% (v/v) glycerol, 0.01% (w/v) bromophenol-blue, and 15% (v/v) β-mercaptoethanol], boiled at 95˚C for 10 min, and centrifuged at 15,000 x g to remove cellular debris. To analyze overall protein levels, lysates were subjected to SDS–PAGE, transferred to polyvinyl difluoride (PVDF) membranes (Pierce Biotechnology, Inc, Waltham, MA), and probed with the appropriate primary antibodies: rabbit-Pglym [1:1000, (*Sullivan, 2003*)], rabbit-Ald, [1:1000, (*Sullivan, 2003*) ], rat-Tm (1:500, Babraham Institute, Cambridge, UK) and mouse anti-ATPase 5α (1:10000, Abcam,

Cambridge, MA). Horseradish Peroxidase (HRP) conjugated secondary antibodies (1:3000–1:5000, GE Healthcare, Chicago, IL) were used to detect primary antibodies. Protein detection was carried out using the ECL Plus western blotting detection kit (ThermoFisher Scientific, Waltham, MA). Densitometry analysis was performed by calculating the relative band intensities of candidate proteins to ATPase 5α loading control using ImageJ software.

## Quantitative PCR (qPCR)

RNA was isolated from a pool of five whole L3 larvae for each genotype using the RNAeasy Mini Kit (QIAGEN, Valencia, CA). After elution, RNA concentrations were determined and single strand complementary DNA (cDNA) was synthesized from 100 ng of RNA using the SuperScript VILO cDNA Synthesis Kit (Invitrogen, Carlsbad, CA). For qPCR, each cDNA sample was diluted to 1:50 and mixed with Power UP SYBR Green Master mix also mixed with the appropriate primers (Applied Biosystems, Foster City, CA). *rp49* was used as the reference gene. Primers were synthesized by Integrated DNA Technologies (IDT):

> *rp49*: F5';-GCCCAAGGGTATCGACAACA-3', R5'-GCGCTTGTTCGATCCGTAAC-3'
> *Ald*: F5'- GGCCGCCGTCTACAAGGC-3', R5'-GTTCTCCTTCTTGCCAGC-3'
> *Pglym78:* F5-AGTCCGAGTGGAACCAGAAGA-3', R5'-GGCTTGAAGTTCTCGTCCAG-3'

Three independent biological replicates were processed for each genotype and reactions were run in triplicate using the Quant Studio 3 Applied Biosystem with Quant studio design and analysis software. The average of the triplicates was used to calculate the 2-ΔΔCt values (normalized fold expression). Quantification of mRNA levels between different genotypes at the same time point was performed using the student t-test.

## Agilent seahorse XFe96 analyzer experiments

*Larval brain, muscle, and wing disc preparation.* The Agilent Seahorse XFe96 metabolic analyzer was placed in an incubator set to 12°C, and analyzer was set to 25°C with the heat on (running Wave software version 2.4). An Agilent Seahorse XFe96 cartridge (Agilent, Santa Clara, CA) was hydrated with 200 µl of calibrant solution (Agilent, Santa Clara, CA) overnight at 25°C. The next day, brains, muscles, or discs were dissected in phosphate buffered solution (PBS) and added to an Agilent 96-well cell plate (Agilent, Santa Clara, CA) containing 50 µl of Agilent Seahorse assay media with supplements required for specific assay (see glycolytic rate assay method). Tissue was sunk to the bottom of the well and centered in the middle between the three raised spheres. Forceps were used to lower the tissue restraint such that the plastic ring is facing toward the bottom of the well and the nylon screen is facing the top of the well. A probe was used to gently push the edge of the tissue restraint down toward the bottom of the well until the restraint did not move or float in the well. 130 µl of assay media was added to each well; resulting in a total of 180 µl final in each well. The cell plate was placed on the tray of the XFe96 analyzer. The instrument was for basal and glycolytic rate assays with all cycle procedures consisting of one-minute mixing, zero-minutes waiting, and 3-min measuring. *Basal ECAR measurements using the XFe96.* Basal levels of extracellular acidification (ECAR) were measured for a minimum of six cycles. Tissue restraints were measured without tissue as a control. Agilent Seahorse XF assay medium (Agilent, Santa Clara, CA) supplemented with 10 mM glucose, and 10 mM sodium pyruvate was used for all basal measurement assays. A minimum of 4 biological replicates were used to analyze the basal rate of *WT* and *tn-/-* larval muscle. Standard error of the mean (SEM) was used in analyzing metabolic measurement levels. Statistical significance was determined using the Holm-Sidak method with alpha = 0.05. *Glycolytic Rate Assay.* Analysis of glycolytic rate with mitochondrial-produced acidification subtracted was conducted using base medium without phenol red (Agilent, Santa Clara, CA), supplemented with 5 mM Hepes (Agilent, Santa Clara, CA), 2 mM glutamine, 10 mM glucose and 1 mM sodium pyruvate. 20 µl of 50 µM rotenone and antimycin-A was added to port A and injected at the 7th cycle, resulting in a final concentration of 5 µM rotenone and antimycin-A. 22 µl of 1M 2-deoxyglutarate (2-DG) was added to port B and injected at the 12th cycle, resulting in a final concentration of 100 mM 2-DG. The software package included with this kit analyzes the oxygen consumption and extracellular acidification rates, while factoring in the buffer capacity of the media. It also calculates the acidification caused by the mitochondria and subtracts this from the data. This method produces the proton efflux rate (PER). A minimum of four biological replicates were used to analyze the glycolytic rate of *WT* and *tn-/-* larval

brains. Standard error of the mean (SEM) was used in analyzing metabolic measurement levels. Statistical significance was determined using the Holm-Sidak method with alpha = 0.05. *Normalization of XFe96 measurements by protein concentration*. XFe96 analysis data was normalized by protein concentration. Protein normalization was conducted using the Pierce 660 nm protein assay reagent (Thermo Fisher Scientific, Waltham, MA). Each sample was homogenized in default lysis buffer (50 mM Tris (pH 7.5), 125 mM NaCl, 5% glycerol, 0.2% IGEPAL, 1.5 mM MgCl$_2$, 1 mM DTT, 25 mM NaF, 1 mM Na$_3$VO$_4$, 1 mM EDTA and 2 $\times$ Complete protease inhibitor (Roche, Indianapolis, IN) on ice, incubated on ice for 15 min, and centrifuged at full speed for 15 min at 4°C. Supernatant was collected and measured as indicated in the Pierce 660 nm protein assay reagent manual. Samples were assayed using a Biotek Cytation three plate reader. PER and muscle ECAR values were divided by µg amount of protein measured to determine the normalized pmol/min/µg rate of proton efflux and mpH/min/µg rate of extracellular acidification. Standard error of the mean (SEM) was used in analyzing metabolic measurement levels. Statistical significance was determined using the Holm-Sidak method with alpha = 0.05.

## Respirometry

Larval metabolic rates were assessed by indirect calorimetry, measuring CO2 production with flow-through respirometry using the Multiple Animal Versatile Energetics platform for metabolic phenotyping (MAVEn, Sable Systems International, Las Vegas, NV). Baseline ultra zero air was provided from a compressed air cylinder and regulated at approximately 20 mL/min through each of 16 respirometry chambers simultaneously. Larvae were measured in groups of five in each chamber. The MAVEn automated flow switching between chambers and baseline (interleave ratio 16:1; dwell time per chamber 2 min). Differential carbon dioxide concentration was measured with a LiCor 7000 infrared gas analyzer. The wet mass of insects prior to and following respirometry was measured to the nearest 0.01 mg (Mettler Toledo XS22SDU Analytical Balance). N = 6 chambers with five larvae in each chamber for each genotype.

## FLIM

The multiphoton based (*Denk et al., 1990*) lifetime and intensity imaging was performed on a custom multiphoton laser scanning system built around an inverted Nikon Eclipse TE2000U at the Laboratory for Optical and Computational Instrumentation (*Yan et al., 2006*). A 20x air immersion objective (Nikon Plan Apo VC, 0.75 NA) (Melville, NY, USA) was used for all imaging. For NADH imaging, data was collected using an excitation wavelength of 740 nm, and the emission was filtered at 457 ± 50 nm (Semrock, Rochester, NY) for the spectral peak for NADH/NADPH. For intensity imaging, the excitation was set at 980 nm, and an emission 520 ± 35 filter was used (Semrock, Rochester, NY). The FLIM fitting process was done according to the methods sections describing FLIM analysis performed for the same scope (*Ghanbari et al., 2019*). For each sample, around eight neighboring fields were randomly selected, and the average value of lifetime and free NADH ratio were calculated.

## L-lactate assays

L-lactate levels in the brain and muscle were measured in the indicated genotypes. L3 larval brains and muscle carcasses were dissected and placed in ice-cold 1X PBS. Brains (n = 20 each, total 60 brains per genotype) or muscle tissue (N = 8, total 24 carcasses per genotype) were pooled together and homogenized in 50 µL 1X PBS. Bradford Assay was used to quantitate protein concentrations. Each lysate was transferred to a 96-well plate and incubated with lactate dehydrogenase and NAD/MTT for 2 hr at room temperature (EnzyChrom Glycolysis Assay Kit, BioAssay Systems). The intensity of the reduced dye was measured at 565 nm, which is directly proportional to the concentration of the L-Lactate in the sample.

## EdU incorporation assay

Dissected larval brains were incubated in *Drosophila* Schneider's medium containing 10 uM EdU (ThermoFisher Scientific, Waltham, MA) for 2 hr at room temperature. Tissues were fixed and Click-iT EdU staining was performed according to the manufacturer's protocol. For quantification, images

of the whole larval brain were captured using a Zeiss 700 confocal microscope. ImageJ was used to manually count the number of EdU positive cells.

## Glucose uptake assay

Wandering L3 larvae of the following genotypes (dpp-Gal4 UAS-mCherry, UAS-LDHGFP >UAS Pvr$^{act}$, tn-/-; dpp-Gal4 UAS-mCherry, UAS-LDHGFP >UAS Pvr$^{act}$) were placed in Drosophila HL3 buffer (NaCl 70 mM, KCl 5 mM, CaCl$_2$1.5 mM, MgCl$_2$20 mM, NaHCO$_3$10 mM, trehalose 5 mM, sucrose 115 mM, and Hepes 5 mM (pH 7.2) on ice. Wing discs were dissected in ice cold PBS followed by incubation with 2-NBDG (2.5 mg/mL, Cayman Chemical, Ann Arbor, MI) for 15 min at room temperature. The tissues were washed with 1X PBS twice, fixed and imaged. Similarly, for analyzing glucose uptake in larval brain, WT and tn-/- mutant brains were dissected and incubated with 2-NBDG as indicated above.

## Food intake assay

Approximately 15–35 early L3 larvae of the indicated genotypes were fed on yeast paste with 0.16% Erioglaucine dye for 24 hr (Aditi et al., 2016). Post feeding larvae were pooled together and washed thrice with distilled water. The pooled larvae were homogenized in 250 µL of ddH$_2$O, centrifuged at 15,000 rpm for 20 min. 225 µL of supernatant was carefully transferred to a 1.5 mL tube containing 50 µL 100% ethanol and centrifuged for 10 min. 250 µL of the supernatant was placed in a new tube and centrifuged again for 5 min. Following centrifugation, 200 µL of the supernatant was placed in a 96-well crystal plate and the OD was measured at 633 nm. Three groups of 15 larvae of each genotype were analyzed.

## TUNEL assay

For TUNEL labeling, fixed brains were treated with 20 µg/mL Proteinase K in PBS for 20 min. For positive control, WT brains were incubated with 100 µl of DNase (1 U/ µl) for 10 min at room temperature post Proteinase K treatment. The tissues were rinsed three times with PBT, followed by a wash in 100 µl terminal deoxynucleotidyltransferase (TdT) equilibration buffer (from the Kit, DeadEnd Fluorometric TUNEL System Promega). For the enzymatic reaction, a working solution of TdT enzyme (30%) with the reaction buffer (70%) was made and added to the tissues. The samples were then incubated for 2 hr in a 37˚C incubator and flickered every 30 min to mix the contents. The reaction was terminated by immersing the tissues in stop buffer (0.3 M NaCl/0.03 M sodium citrate) for 15 min at room temperature. DAPI (1:400) was used as a counterstain.

## Acknowledgements

We thank Jim Vigoreaux for sharing antibodies and Kasra X Ramyar, Joe McWhorter, and Samantha Gameros for technical assistance. We also thank James Cox for services at the University of Utah Metabolomics Core as well as Steve Hartson and Janet Rogers for services and guidance at the Oklahoma State University Proteomics and Mass Spectrometry Core. Special appreciation to the VDRC and BDSC for fly lines used in this study. This work was supported by grant through the National Institute of Arthritis and Musculoskeletal and Skin Diseases (NIAMS) to ERG (R21AR073373, R01AR060788, and R56AR060788). JMT is supported by a MIRA award from NIGMS (R35GM119557). X-ray diffraction data were collected at Southeast Regional Collaborative Access Team 22-BM beamline at the Advanced Photon Source, Argonne National Laboratory. Supporting institutions may be found at www.ser-cat.org/members.html. Use of the Advanced Photon Source was supported by the US Department of Energy, Office of Science, Office of Basic Energy Sciences, under Contract W-31–109-Eng-38*.

# Additional information

## Funding

| Funder | Grant reference number | Author |
| --- | --- | --- |
| National Institute of Arthritis and Musculoskeletal and Skin Diseases | R01AR060788 | Erika R Geisbrecht |
| National Institute of General Medical Sciences | R35GM119557 | Jason M Tennessen |
| National Institute of Arthritis and Musculoskeletal and Skin Diseases | R21AR073373 | Erika R Geisbrecht |

The funders had no role in study design, data collection and interpretation, or the decision to submit the work for publication.

## Author contributions

Simranjot Bawa, Conceptualization, Formal analysis, Investigation, Writing - review and editing; David S Brooks, Resources, Investigation; Kathryn E Neville, Md Abdul Sagar, Joseph A Kollhoff, Kevin W Eliceiri, Investigation; Marla Tipping, Brian V Geisbrecht, Formal analysis, Investigation; Geetanjali Chawla, Resources; Jason M Tennessen, Formal analysis, Writing - review and editing; Erika R Geisbrecht, Conceptualization, Formal analysis, Supervision, Funding acquisition, Writing - original draft, Writing - review and editing

## Author ORCIDs

Simranjot Bawa (iD) http://orcid.org/0000-0003-3837-3868
Md Abdul Sagar (iD) http://orcid.org/0000-0002-1564-0727
Jason M Tennessen (iD) http://orcid.org/0000-0002-3527-5683
Kevin W Eliceiri (iD) http://orcid.org/0000-0001-8678-670X
Erika R Geisbrecht (iD) https://orcid.org/0000-0002-1450-7166

## Decision letter and Author response

Decision letter https://doi.org/10.7554/eLife.52358.sa1
Author response https://doi.org/10.7554/eLife.52358.sa2

# Additional files

## Supplementary files

• Transparent reporting form

## Data availability

Diffraction data have been deposited in PDB under the accession code 6D69. All other data generated or analysed during this study are included in the manuscript and supporting files. Source data files have been provided for all raw data.

The following dataset was generated:

| Author(s) | Year | Dataset title | Dataset URL | Database and Identifier |
| --- | --- | --- | --- | --- |
| Ramyarn KX, McWhorter WJ, Geisbrecht BV | 2018 | Crystal Structure of the NHL Repeat Region of D. melanogaster Thin | https://www.rcsb.org/structure/6D69 | RCSB Protein Data Bank, 6D69 |

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
