## [Decision Letter]

**Acceptance summary:**

Cells undergo a constant shift between catabolism and anabolism to meet the metabolic needs of their growth. Biochemical and genetic studies using *Drosophila* provide critical insights into how cells achieve such a switch via the E3 ligase protein, TRIM32. In addition to its role in protein degradation, authors have shown that TRIM32 physically interacts with and stabilizes key glycolytic proteins, Aldolase and Phosphoglycerate mutase and controls glycolytic flux. Given that its function is proven in both the muscle and in overgrowing tissues, it will be interesting to see detailed molecular mechanisms of TRIM32 that may be conserved in other species.

**Decision letter after peer review:**

Thank you for submitting your article "*Drosophila* TRIM32 cooperates with glycolytic enzymes to promote cell growth" for consideration by *eLife*. Your article has been reviewed by three peer reviewers, one of whom is a member of our Board of Reviewing Editors, and the evaluation has been overseen by Utpal Banerjee as the Senior Editor. The reviewers have opted to remain anonymous.

The reviewers have discussed the reviews with one another and the Reviewing Editor has drafted this decision to help you prepare a revised submission.

As you will see, all of the reviewers agreed on the significance of your work, but a number of critical criticisms were raised which require new experimental data and analyses. The full comments of the reviewers are attached to provide further details.

Summary:

In this study, authors have discovered a novel role for TRIM32 in the glycolytic control mediated by interactions with Aldolase and Pglym, enzymes responsible for the glycolytic pathway. Loss of TRIM32 leads to an overall loss of glycolytic flux and related metabolic parameters in the muscle. Furthermore, TRIM32 is required for the Pvr-induced tumor growth, demonstrating that TRIM32 is a critical regulator of glycolytic enzymes in both developing and cancer tissues. Overall, this study will be of interest to a broad audience studying metabolism, cancer, muscle biology, and the TRIM32 associated diseases, prompting future works.

Essential revisions:

1) The authors need to demonstrate whether the *tn* mutant phenotype is cell-autonomous.

– Tissue-specific knock-down or *tn*-/- mutant clones in the muscle/brain will address the concerns.

– Quantifying food intake by *tn* mutant larvae will resolve a part of the issue.

2) Related to concern #1, there is only one *tn* mutant allele used in this study. It will be critical to show phenotypes of another *tn* mutant allele or multiple RNAi lines.

3) Interactions between TRIM32 and Ald or Pglym were only shown in vitro. In vivo evidence of TRIM32-Ald or TRIM32-Pglym interaction will be needed.

4) The metabolic phenotypes require further characterization including:

– Respiration in *tn* mutants

– Quantification/analysis or better explanations on glycolytic flux, glucose uptake and ATP production

– Quantify glycolytic flux rates in the wing disc

– Compare PER in *WT* and *tn* muscles

5) Clarify the causality of the glycolytic defect and the muscle phenotypes.

The reviewers' comments attached below are detailed and will help you improve the manuscript for publication.

Reviewer #1:

In this study, Geisbrecht and authors have identified a novel role for TRIM32 in the control of glycolytic flux by physical interactions with Aldolase and Pglym. Authors have characterized structural homologies between the mammalian TRIM32 and *Drosophila* thin, and isolated glycolytic enzymes including Aldolase and Pglym as biochemical interactors. Loss of TRIM32 leads to an overall loss of glycolytic metabolites and 11 amino acids, and as a consequence, ATP levels significantly reduce in this background. Furthermore, authors have shown TRIM32 is required for the Pvr-induced tumor growth in the wing disc, concluding that TRIM32 is a critical regulator of glycolytic enzymes in both developing and cancerous tissues.

1) Authors have indicated that *tn-/-* larvae significantly decreased ATP levels as well as glycolytic products. These metabolic consequences could be due to reduced food consumption or dysfunctions in the digestive system. Authors need to distinguish systemic developmental defects and cell-autonomous functions of *tn* mutants. It will be critical to include proper controls showing that the metabolic phenotype is not derived by *tn*-mediated systemic effects but by loss of *tn* in a specific tissue. As one of the examples, tissue-specific metabolic measurements after the loss of *tn* in the muscle or the brain could suitably support the query.

2) The authors' previous study on *tn* mutants has shown that loss of *tn* causes an overall reduction in the muscle and animal size. In Figure 4A, authors have claimed that *tn* mutants exhibit the significantly smaller size of the brain, in addition to the muscle. However, given that *tn* mutants are smaller in their size, it will be important to show proportional size changes of the brain and the muscle compared to the larval size changes.

3) According to the Materials and methods, authors cultured *WT* or *tn* mutant larvae on the agar plate from very early stages and let them grow until the 3rd instar. However, *WT* animals marginally change their body mass and the muscle diameter after the chronic starvation (Figure 3F-I). How do larvae properly grow without any nutrition? Supporting references on the chronic starvation and normal growth, or some other control experiments would make the data more concrete and convincing.

4) If NHL domain plays a unique role in protecting the glycolytic enzymes, a specific deletion or mutation on the NHL domain would provide more precise metabolic phenotypes while eliminating Ub-mediated complexities. It is not clear in the current version whether it is the NHL domain mutation that gives rise to the glycolytic phenotypes. It will be important to segregate non-canonical TRIM32 functions through NHL domain from Ub-mediated canonical phenotypes.

5) In Figure 4E-H, authors have shown that the Pvr-induced tumor growth phenotype is recovered by loss of *tn*. Though the representative image in Figure 4F displays significantly bigger wing disc, the quantitation of the *dpp>Pvr^act^* shown in Figure 4H indicates a comparable measurement to that of controls. Accurate measurement would enhance the clarity of data.

Reviewer #2:

This is an interesting study uncovering a novel function for the disease-associated protein TRIM32. This will likely be of interest to a broad audience studying metabolism, muscle biology, as well as the TRIM32-associated diseases.

A number of issues need to be significantly strengthened, however, to make the main conclusions of this study solid:

1) Does full-length TRIM32 bind Ald and Pglym in vivo? The authors only show an in vitro binding assay using recombinant proteins (likely in high molar concentrations) using a truncated TRIM32 version. Can these protein-protein interactions be observed by co-immunoprecipitation using full-length TRIM32 from tissue or cell lysates?

2) It is not clear that the phenotypes described here are indeed cell-autonomous, as would be expected if they are due to glycolytic defects in the cells being studied. Since whole-body *tn* mutants are being studied, the phenotypes could be due to more complex organismal defects. Specifically, are the reduced size of muscle and brain cells cell-autonomous? If *tn* is knocked-down only in the brain or only in a small subset of larval muscles, or if *tn*- mutant clones are generated in the brain, does this result in small cell size?

3) The metabolic phenotype is not sufficiently characterized:

– Figure 2: A decrease in lactate production does not mean glycolytic flux is reduced – it could also indicate an increase in respiration. The drop in steady-state pyruvate levels is not interpretable because it does not say anything about flux through pyruvate. (i.e. if twice as much pyruvate is made per unit of time, and twice as much pyruvate is used up, steady-state pyruvate levels will not change, showing that they do not say anything about flux). Is respiration increased in *tn* mutant cells?

– It is highly unlikely, if indeed glycolytic flux is reduced, that there is no change in glucose uptake by these cells, as concluded in the manuscript, referring to Figure 4—figure supplement 2. Almost always, changes in glycolytic flux and glucose uptake correlate. Where would all the intracellular glucose go otherwise? Even if it shuttles into the PPP pathway it still returns to the glycolytic pathway as glyceraldehyde 3-phosphate and fructose 6-phosphate which requires Pglym to be metabolized… Hence Figure 4—figure supplement 2E-F would need to be quantified (e.g. cell dissociation and FACS? Or lysis and measuring fluorescence normalized to protein?) to conclude this more robustly.

– The authors write "The metabolism of growing cells must strike a balance between ATP production and the maintenance of metabolite pools that contribute to biomass production. The metabolomic profile of *tn*-/- larvae suggests that TRIM32 regulates this metabolic balance by promoting glycolytic flux, which results in the synthesis (and preservation) of amino acid pools for protein synthesis."

This conclusion does not appear to be correct. Indeed, more respiration will lead to more ATP synthesis and less biomass production. However, the *tn* mutants do not have less lactate and more ATP, but less of everything (less lactate, less ATP and less amino acids).

– Is TRIM32 expressed in cells in culture (e.g. *Drosophila* S2 cells)? If so, these issues can be dissected much more easily in cell culture, but knocking down TRIM32 and quantitating glucose uptake, lactate production, and oxygen consumption. This would more rigorously show an effect on glycolytic flux.

4) It is not clear that indeed a glycolytic defect is causing the muscle phenotypes.

The authors write "To confirm that the smaller muscles in TRIM32-deficient larvae are indeed due to defective glycolysis, muscle carcasses were isolated and assessed for changes in metabolic activity" – this statement is misleading/incorrect. This approach only shows correlation, not causality.

– There is no functional experiment in the manuscript showing that reduced glycolysis is causing the muscle phenotypes, or that restoring glycolytic flux rescues the phenotype.

5) Importantly, from Figure 3F, the main conclusion that "the addition of amino acid building blocks is sufficient to rescue a TRIM32-mediated loss of muscle mass." is not warranted.

The addition of yeast extract or amino acids does not 'rescue' the muscle defect of *tn* mutants, i.e. it does not return the phenotype to wildtype levels. It simply rescues the additional defect caused by not feeding these animals any protein (agar only). This does not mean much. It is analogous to not feeding *tn* mutants any water and seeing that they dehydrate. Then giving them water again would return them to the original *tn* mutant phenotype, and thereby concluding that water is sufficient to rescue the TRIM32-mediated phenotype.

To prove that amino acid deficiency is causing the muscle atrophy phenotype, one would need to see that feeding additional amino acids (on top of the normal food) to *tn* mutants causes their muscle size to become more like wildtype.

6) Could it be the other way around – that a muscle defect is leading to an eating defect (which requires continuous muscle movement) and as a consequence, less intake of sugar and protein by the larvae, leading to the metabolic phenotypes? This can be resolved by:

– quantifying food intake by *tn* mutant larvae;

–- checking the cell-autonomy of the metabolic defects;

– testing in cell culture whether TRIM32 knockdown leads to the same glycolytic defects.

7) Figure 4—figure supplement 1: it is difficult to judge whether apoptosis rates account for a difference in tissue size by just quantifying apoptosis, because this only yields a snapshot of apoptosis at that moment, whereas tissue size integrates apoptosis rates over time. So a change in apoptosis may appear mild at any one point, but result in significant tissue size differences over time. The way to prove this functionally is to block apoptosis, e.g. with p35. Alternatively, if the authors want to conclude that cell size in *tn* brains is smaller than in control brains, a direct quantification of cell size would be better.

8) The authors write "Wing discs are not inherently glycolytic and do not endogenously express LDH (Figure 4E). Accordingly, the overall area or volume of the disc was not affected by loss of TRIM32"

This is not correct. Endogenous LDH (called ImpL3 in *Drosophila*) is expressed in wing discs, and can be detected by Q-RT-PCR or by in situ hybridization on endogenous transcript.

Hence, the fact that wing disc size is not decreased in TRIM32 mutants cannot be explained with this justification and raises the question whether indeed the metabolic mechanism the authors propose is correct.

9) Along the same lines, there is no quantification of glycolytic flux rates in the wing discs – neither that PVR expression causes them to increase, nor that the *tn* mutation causes them to drop again. The phenotype shown in Figure 4H simply says there is some genetic interaction, but this could be happening at any level.

In sum, between points #4 and #5, I do not see any solid evidence that a metabolic defect is causing the muscle phenotypes described here.

Reviewer #3:

Bawa et al. describes compelling observations suggesting that TRIM32 is a key regulator of glycolysis in fast growing larval tissues. *Drosophila* TRIM32 ortholog Tn physically interacts with two enzymes, Aldolase and Pglym. Strikingly, loss of *tn* caused a reduction in Ald and Pglym protein levels, which might not be explained by its role as E3-ubiquitin ligase. Of importance, the authors present multiple pieces of data suggesting that Tn is required for maintaining the levels of metabolites produced by glycolysis and derived from glucose. Biochemical and physiological characterization of *tn* mutant shows that a main role of *tn* is to maintain amino acid pools to support growth presumably by maintaining glycolysis. Overall, the observations are quite interesting and novel, prompting follow-up studies to elucidate the molecular mechanisms by which Tn maintains glycolysis and address whether mammalian TRIM32 plays a similar role. The manuscript should be suitable for publication if the authors address the following comments.

1) Proton efflux rate (PER), as shown with brain tissues in Figure 4D, provides a reasonable assessment of glycolytic flux in isolated tissues. In contrast, extracellular acidification rate (ECAR) is also influenced by CO_2_ produced during respiration; thus it wouldn't provide an accurate estimation of glycolytic flux. To clearly demonstrate *Drosophila* TRIM32 regulate glycolysis in larval muscle, it is critical to compare PER in wild-type and *tn* muscles.

2) Only one *tn* mutant allele is used throughout the study. How can the authors rule out the possibility that the phenotypes observed with the *tn* allele is not due to a background mutation? One way to resolve this issue is to show that the key phenotypes, such as a reduction in PER, can be reversed by having a genomic rescue transgene or tissue specific rescue using GAL4/UAS system. Alternatively, it would be helpful to show that similar phenotypes can be seen with different *tn* alleles.

3) Considering the hypothesis that TRIM32 is required for maintaining glycolytic flux, I would suggest to move the results in Figure 2—figure supplement 2C-D to the main figures. Note that the results shown in Figure 2C-G are not new; additionally, similar results are shown in Figure 3B-F. Thus, Figure 2C-G can be moved to the supplementary figures.

[Editors' note: further revisions were suggested prior to acceptance, as described below.]

Thank you for resubmitting your work entitled "*Drosophila* TRIM32 cooperates with glycolytic enzymes to promote cell growth" for further consideration by *eLife*. Your revised article has been evaluated by Utpal Banerjee as the Senior Editor, and a Reviewing Editor.

The manuscript has been improved but there are some remaining issues that need to be addressed before acceptance, as outlined below:

1) As suggested by reviewer #1 comment 2, reviewer #2 comment 1-3, in vivo interactions between TRIM32 and Ald/Pglym are critical to prove the hypothesis, and therefore, require precise descriptions of experiments and results.

2) Reviewer #3 comment 2: ERR-mediated restoration experiments need better descriptions and if possible, additional data could be added.

3) Other comments can be immediately addressed or edited.

Reviewer #1:

Authors have significantly improved the manuscript and adequately addressed critiques raised by reviewers. There are some points below to be changed for better readability.

1) In the line "Importantly, protein levels of Ald.…… ERR in *tn-/-* muscles, indicating that~", figure citation is missing. It could be Figure 2—figure supplement 2F-G.

2) As suggested in the major revision 3, showing the in vivo interactions between TRIM32 and Ald/Pglym is essential for the study. Therefore, Author response image 1 needs to be moved to the main figure with additional explanations and discussions.

3) Given that *tn* mutants exhibit reduced mouth hook contraction, the growth of *tn* mutants could be overall retarded while tissue-specific knockdown of *tn* showed no difference. Having the possibility of slow growth, showing the small-sized muscle or brain of *tn* mutants in main Figures 3-4 could be misleading. It will be more adequate to show the small size phenotypes with tissue-specific RNAi backgrounds.

Reviewer #2:

The manuscript is significantly improved. For instance, the authors do a good job of showing the tissue-autonomous nature of the TRIM32 phenotypes.

Some issues which can be fixed without any additional experiments:

1) The authors write in their rebuttal "Identification of the TRIM32-Ald or TRIM32-Pglym biochemical interaction was performed in vivo using larval lysates (Figure 1C; Figure 1—figure supplement 2A…". If I understand correctly, in Figure 1, a truncated version of TRIM32 was fused to GST, expressed recombinantly in bacteria, and purified. This was then incubated in a test tube with larval lysates. How can this be called "in vivo"? Some people call cells in culture "in vivo" and some people mean a live organism by "in vivo", but I do not think anyone considers a lysate in a test tube to be "in vivo"? If I misunderstood the experimental setup, it needs to be described better in the manuscript.

2) Showing that full-length TRIM32 binds Ald and Pglym in vivo is critical for this story. I believe the results shown in Author response image 1 should be put into the manuscript.

3) Regarding the fact that the co-IPed Ald and Pglym in Author response image 1 are running at a different molecular weight compared to Ald and Pglym in the input can either be interpreted as the authors write by post-translational modification of the pool of Ald and Pglym that interact with TRIM32, or it might simply be another contaminating protein that cross-reacts with the Ald and Pglym antibodies. Can this be distinguished? If it is a sub-population of the total Ald/Pglym, it should also be visible in the total lysates. Also, most post-translational modifications are readily lost in IP buffer and Laemmli (e.g. ubiquitination or phosphorylation), which would cause it to run at the 'normal' size. This should be discussed.

4) The authors write "Surprisingly, cellular glucose uptake… was normal in *tn-/-* larval brains and wing discs… demonstrating that glucose is not a limiting substrate for glycolysis in these isolated tissues."

I believe this statement is not correct. How does this show glucose is not a limiting substrate for glycolysis in brains and wing discs? Or do the authors mean specifically in *tn* mutants?

Reviewer #3:

The authors have appropriately addressed most of my comments. I have two additional comments:

1) The authors showed that Ald and Pglym protein levels were reduced in *tn* mutant. However, the authors didn't show that TRIM32 stabilized these proteins. Thus, I suggest to revise the following sentences:

"Here we provide a novel mechanism for TRIM32 in cell growth. Our data show that TRIM32 promotes glucose metabolism through the stabilization of glycolytic enzyme levels."

"Taken together, these data highlight a unique role for TRIM32 in the stabilization of glycolytic enzyme levels."

"Importantly, protein levels of Ald and Pglym were stabilized upon expression of ERR in *tn*-/- muscles, indicating that restoration of glycolytic protein levels is sufficient to recover TRIM32-mediated growth defects."

2) Although the authors claimed that ERR overexpression stabilized Ald and Pglym levels, the results were not shown. Thus, the authors should test whether ERR overexpression could restore Ald and Pglym levels in *tn* mutant. Otherwise, the result regarding the ERR overexpression in muscle shown in Figure 2M is better to be removed. Additionally, given that ERR can impact on multiple targets, "restoration of glycolytic protein levels is sufficient to recover TRIM32-mediated growth defects." is an overstatement.

---

## [Author Response]

Essential revisions:1) The authors need to demonstrate whether the tn mutant phenotype is cell-autonomous.– Tissue-specific knock-down or tn-/- mutant clones in the muscle/brain will address the concerns.

This is a great point. Multiple new experiments show that the *tn* mutant phenotype is cell-autonomous:

1) Knockdown of TRIM32 using three independent RNAi lines all show a reduction in the diameter of L3 muscles upon induction with *mef2-*Gal4 (*tn RNAi-A, tn RNAi-B*, and *tn RNAi-C*) (Figure 2K; Figure 2—figure supplement 2A, B). Note that two of these RNA lines were previously published by us (LaBeau-DiMenna et al., 2012) and the Nguyen group (Domsch et al., 2013) to show the same small muscle phenotype as the *tn* alleles in this paper. We also show that knockdown of *tn RNAi-A* in a single muscle (*5053*-Gal4 driver) reduces cell size (Figure 2—figure supplement 2C).

2) Brain-specific knockdown of TRIM32 using the pan-neuronal driver *elav*-Gal4 causes a smaller larval brain size, while knockdown of *tn RNAi* in muscles does not alter brain size (Figure 4C-G).

3) Brain lactate levels are lower in *elav>tn RNAi*, but not *mef2>tn RNAi* brains (Figure 4I). The converse is also true, where muscle carcasses of the genotype *mef2>tn RNAi* show a reduction in lactate levels, but not upon neuronal-specific depletion of TRIM32 (*elav>tn RNAi*) (Figure 2L).

4) While previously published (LaBeau-DiMenna et al., 2012; Domsch et al., 2013), we confirm that muscle-specific expression of *TRIM32 cDNA* rescues muscle cell size (Figure 2M), muscle function (Figure 2—figure supplement 2D, E), and stabilizes glycolytic enzyme levels (Figure 2—figure supplement 2F, G).

5) Finally, two tissues that are not considered glycolytic in nature (wing disc and midgut), are not smaller in *tn* mutants. This is further evidence that TRIM32 affects glycolytic tissues (muscle and brain).

– Quantifying food intake by tn mutant larvae will resolve a part of the issue.

As suggested by multiple reviewers, muscle defects could lead to reduced feeding and thus produce whole body metabolic defects. To assay food intake, larvae were fed yeast mixed with a blue dye (Aditi et al., 2016). The amount of dye present in the gut of *WT, tn-/-,* or *tn RNAi* animals was quantified by measuring absorbance values at a wavelength of 600 nm. Not surprisingly, *tn-/-* larvae showed reduced mouth hook contractions and compromised ingestion of dye at both 3 and 24 hr (Figure 5—figure supplement 2E, I, J). However, larvae with reduced TRIM32 in larval muscle (*mef2>tn RNAi*) or brain tissue (*elav>tn RNAi*) showed no difference between control or RNAi knockdown conditions (Figure 5—figure supplement 2I, J). These data clearly demonstrate that a loss of TRIM32 in all tissues compromises food consumption, while the observed metabolic defects (e.g., lactate production) upon tissue-specific loss of TRIM32 are not nutrition limited.

2) Related to concern #1, there is only one tn mutant allele used in this study. It will be critical to show phenotypes of another tn mutant allele or multiple RNAi lines.

We should have emphasized this point in our initial submission. Our 2012 PNAS paper described three *tn* alleles [*l(2)tn, tnΔA*, and *tnΔB*] and one RNAi line (*tn RNAi-A*) that exhibit smaller, degenerative muscles (LaBeau-DiMenna et al., 2012). Soon after, the Nguyen lab published similar results using three additional alleles of *tn* as well as the *tn RNAi-A* and *tn RNAi-B* lines used in this paper (Domsch et al., 2013). Note that a later publication from our lab confirmed knockdown of *tn mRNA* transcripts for two *UAS-tn RNAi* lines by qPCR (Brooks et al., 2016). To further support our results, we now show in this manuscript that knockdown of three RNAi lines (*tn RNAi-A, tn RNAi-B,* and *tn RNAi-C*) in muscle tissue produce smaller muscles (Figure 2K) and also exhibit reduced lactate levels (Figure 2L).

3) Interactions between TRIM32 and Ald or Pglym were only shown in vitro. In vivo evidence of TRIM32-Ald or TRIM32-Pglym interaction will be needed.

Identification of the TRIM32-Ald or TRIM32-Pglym biochemical interaction was performed in vivo using larval lysates (Figure 1C; Figure 1—figure supplement 2A) and we later developed the in vitro assay to validate these in vivo interactions (Figure 1D, E; Figure 1—figure supplement 2B). Nevertheless, to address reviewer concerns, we performed immunoprecipitations using anti-TRIM32 antibodies to pull down protein complexes from L3 larval lysates. As shown in Author response image 1, we can pulldown TM, Ald, and Pglym and uncovered quite an interesting result. While the molecular weight of immunoprecipitated TM is as expected, we see higher molecular weight bands of Ald and Pglym in our anti-TRIM32 IPs that are suggestive of a post-translational modification (PTM). This verification of TRIM32-glycolytic enzyme complexes will be followed up to identify the nature of the PTM. We include a section at the end of the Discussion entitled ‘Limitations of this study’ that addresses how TRIM32 may function with glycolytic enzymes.

**Author response image 1. respfig1:** Reviewer Figure 1.

4) The metabolic phenotypes require further characterization including:– Respiration in tn mutants

In Figure 2—figure supplement 1B, we show that CO_2_ production is mildly increased in *tn-/-* whole larvae. Given that multiple tissues with metabolic phenotypes may contribute to CO_2_ production, we further assessed PER in isolated muscle (Figure 2I) and brain (Figure 4H) tissue. These results confirm a tissue-specific decrease in glycolytic activity.

– Quantification/analysis or better explanations on glycolytic flux, glucose uptake and ATP production

As this is a short report, we limited text explanations in our original submission. In this revised version, the text has been extensively modified to improve readability. We have added multiple experiments (and associated analysis/explanations) to better assess glycolytic flux: (1) PER in isolated muscle carcasses (Figure 2I); (2) PER in wing discs (Figure 5—figure supplement 1D); (3) FLIM analysis in control and tumorous wing discs (Figure 5F, G).

– Quantify glycolytic flux rates in the wing disc

We now show that glycolytic flux rates, assessed by PER (Figure 5—figure supplement 1D) and FILM (Figure 5F, G), as well as overall wing disc size (Figure 5D, Figure 5—figure supplement 1C) are not different between *WT* and *tn-/-* wing discs. This negative result is very important as it demonstrates that the observed decreases in cell size due to loss of TRIM32 only affects highly glycolytic tissues for maximizing substrate production during rapid cell growth.

– Compare PER in WT and tn muscles

We now include PER data for *WT* and *tn-/-* muscles (Figure 2I). Our results show that glycolytic flux is reduced in TRIM32 mutant muscle tissue.

5) Clarify the causality of the glycolytic defect and the muscle phenotypes.

To confirm that reduced glycolysis is indeed causing the reduction in muscle size, we expressed Estrogen-related receptor (ERR) in muscle tissue. ERR is a nuclear hormone receptor that acts as a transcriptional switch in embryogenesis to induce genes required for aerobic glycolysis during larval growth (Tennessen et al., 2011). Indeed, muscle-specific expression of ERR in *tn-/-* restores muscle size (Figure 2M), allows for muscle contraction (Figure 2—figure supplement 2D, E), and stabilizes the levels of Ald and Pglym (Figure 2—figure supplement 2F, G).

The reviewers' comments attached below are detailed and will help you improve the manuscript for publication.Reviewer #1:[…]1) Authors have indicated that tn-/- larvae significantly decreased ATP levels as well as glycolytic products. These metabolic consequences could be due to reduced food consumption or dysfunctions in the digestive system. Authors need to distinguish systemic developmental defects and cell-autonomous functions of tn mutants. It will be critical to include proper controls showing that the metabolic phenotype is not derived by tn-mediated systemic effects but by loss of tn in a specific tissue. As one of the examples, tissue-specific metabolic measurements after the loss of tn in the muscle or the brain could suitably support the query.

Please see complete response above (Essential revisions #1). In summary, we now show that muscle-specific or brain-specific loss of TRIM32 causes a reduction in L-lactate levels, independent of feed consumption.

2) The authors' previous study on tn mutants has shown that loss of tn causes an overall reduction in the muscle and animal size. In Figure 4A, authors have claimed that tn mutants exhibit the significantly smaller size of the brain, in addition to the muscle. However, given that tn mutants are smaller in their size, it will be important to show proportional size changes of the brain and the muscle compared to the larval size changes.

We previously showed that *tn* mutants cause a reduction in muscle cell size, but not overall animal size (a phenotype associated with growth defects) (LaBeau-DiMenna, et al. 2012). In fact, loss of TRIM32 results in elongated pupal cases due to defective muscle contraction during the larval to pupal transition (Figure 3H, I; Figure 2—figure supplement 2D, E). Thus, the smaller muscle and brain sizes affect only these tissues, but not the size of wing discs (Figure 5D; Figure 5—figure supplement 1A-C), midgut tissue (Figure 5—figure supplement 2F-H), or overall body size.

3) According to the Materials and methods, authors cultured WT or tn mutant larvae on the agar plate from very early stages and let them grow until the 3rd instar. However, WT animals marginally change their body mass and the muscle diameter after the chronic starvation (Figure 3F-I). How do larvae properly grow without any nutrition? Supporting references on the chronic starvation and normal growth, or some other control experiments would make the data more concrete and convincing.

Thank you for bringing this to our attention as we should have explained this more clearly in our initial submission. The reviewer is correct that complete starvation during early larval development causes growth arrest and reduced size (Beadle et al., 1938; Robertson, 1966). Our larvae were reared on 2.25% agar, which contains an undefined amount of two polysaccharides, agarose and agaropectin. Thus, the larvae are not devoid of all nutrition. In our hands, *WT* larval growth is delayed with variable L3 body mass and muscle diameter (Figure 3F, G). Some *WT* larvae also die, but others are able to pupate, albeit reduced in size (Figure 3H, I). We have clarified this information in Materials and methods.

4) If NHL domain plays a unique role in protecting the glycolytic enzymes, a specific deletion or mutation on the NHL domain would provide more precise metabolic phenotypes while eliminating Ub-mediated complexities. It is not clear in the current version whether it is the NHL domain mutation that gives rise to the glycolytic phenotypes. It will be important to segregate non-canonical TRIM32 functions through NHL domain from Ub-mediated canonical phenotypes.

Yes! We agree these are important experiments. Preliminary results in our lab show that deletion of the TRIM32 NHL domain fails to rescue *tn-/-* phenotypes in muscle (also published in Domsch et al., 2013). Experiments are underway to examine the necessity of human point mutations for maintaining glycolysis in muscles.

5) In Figure 4E-H, authors have shown that the Pvr-induced tumor growth phenotype is recovered by loss of tn. Though the representative image in Figure 4F displays significantly bigger wing disc, the quantitation of the dpp>Pvr^act^ shown in Figure 4H indicates a comparable measurement to that of controls. Accurate measurement would enhance the clarity of data.

There is indeed variability in wing disc area of *dpp>Pvr^act^* tumors and measuring volume appears to better assess overall size (previously supplementary data). We now show volume measurements in Figure 5D and moved area data for wing discs to Figure 5—figure supplement 1C. Note there is a clear difference between tumors grown in *WT* or *tn-/-* backgrounds when area is measured.

Reviewer #2:This is an interesting study uncovering a novel function for the disease-associated protein TRIM32. This will likely be of interest to a broad audience studying metabolism, muscle biology, as well as the TRIM32-associated diseases.A number of issues need to be significantly strengthened, however, to make the main conclusions of this study solid:1) Does full-length TRIM32 bind Ald and Pglym in vivo? The authors only show an in vitro binding assay using recombinant proteins (likely in high molar concentrations) using a truncated TRIM32 version. Can these protein-protein interactions be observed by co-immunoprecipitation using full-length TRIM32 from tissue or cell lysates?

Please see our response above (Essential revisions #3) and Author response image 1.

2) It is not clear that the phenotypes described here are indeed cell-autonomous, as would be expected if they are due to glycolytic defects in the cells being studied. Since whole-body tn mutants are being studied, the phenotypes could be due to more complex organismal defects. Specifically, are the reduced size of muscle and brain cells cell-autonomous ? If tn is knocked-down only in the brain or only in a small subset of larval muscles, or if tn- mutant clones are generated in the brain, does this result in small cell size?

This data is now included (Figure 2M; Figure 2—figure supplement 2A-C; Figure 2L; Figure 4C-G; Figure 4I). Please see previous responses (Essential revisions #1 and reviewer #1, point #1).

3) The metabolic phenotype is not sufficiently characterized:– Figure 2: A decrease in lactate production does not mean glycolytic flux is reduced – it could also indicate an increase in respiration. The drop in steady-state pyruvate levels is not interpretable because it does not say anything about flux through pyruvate. (i.e. if twice as much pyruvate is made per unit of time, and twice as much pyruvate is used up, steady-state pyruvate levels will not change, showing that they do not say anything about flux). Is respiration increased in tn mutant cells?

This is a great point. This data is now included (Figure 2—figure supplement 2B). Please see responses above (Essential revisions #4) and the revised text.

– It is highly unlikely, if indeed glycolytic flux is reduced, that there is no change in glucose uptake by these cells, as concluded in the manuscript, referring to Figure 4—figure supplement 2. Almost always, changes in glycolytic flux and glucose uptake correlate. Where would all the intracellular glucose go otherwise? Even if it shuttles into the PPP pathway it still returns to the glycolytic pathway as glyceraldehyde 3-phosphate and fructose 6-phosphate which requires Pglym to be metabolized… Hence Figure 4—figure supplement 2E-F would need to be quantified (e.g. cell dissociation and FACS? Or lysis and measuring fluorescence normalized to protein?) to conclude this more robustly.

This result is surprising to us as well. Since we did not look at glucose uptake in other tissues or quantify differences using cell dissociation and FACS, it is possible there are minor changes that we were not able to observe. Nevertheless, we can conclude that the tissue-specific reductions in cell size are independent of nutrient status (Figure 5—figure supplement A-J).

– The authors write "The metabolism of growing cells must strike a balance between ATP production and the maintenance of metabolite pools that contribute to biomass production. The metabolomic profile of tn-/- larvae suggests that TRIM32 regulates this metabolic balance by promoting glycolytic flux, which results in the synthesis (and preservation) of amino acid pools for protein synthesis."This conclusion does not appear to be correct. Indeed, more respiration will lead to more ATP synthesis and less biomass production. However, the tn mutants do not have less lactate and more ATP, but less of everything (less lactate, less ATP and less amino acids).

We have removed these sentences entirely.

– Is TRIM32 expressed in cells in culture (e.g. Drosophila S2 cells)? If so, these issues can be dissected much more easily in cell culture, but knocking down TRIM32 and quantitating glucose uptake, lactate production, and oxygen consumption. This would more rigorously show an effect on glycolytic flux.

We agree that cell culture can sometimes be an easier system to dissect cellular mechanisms, but we feel it is not ideal for our current studies. *Drosophila* S2 cells are not inherently glycolytic, although stress conditions can induce a shift in glucose metabolism from oxidative phosphorylation to glycolysis (Freije et al., 2012; Lee et al., 2015).

4) It is not clear that indeed a glycolytic defect is causing the muscle phenotypes.The authors write "To confirm that the smaller muscles in TRIM32-deficient larvae are indeed due to defective glycolysis, muscle carcasses were isolated and assessed for changes in metabolic activity" – this statement is misleading/incorrect. This approach only shows correlation, not causality.

This statement has been deleted due to massive text revisions. See also previous responses (Essential revisions #1, #3, #5) as we now prove that a decrease in glycolytic activity causes smaller muscle size.

– There is no functional experiment in the manuscript showing that reduced glycolysis is causing the muscle phenotypes, or that restoring glycolytic flux rescues the phenotype.

This is a great suggestion. As we detailed above (Essential revisions #5), we now show that induction of carbohydrate metabolism genes by ERR restores glycolytic protein levels and muscle mass. These results provide strong evidence for a causal link between glycolysis and the accumulation of biomass in muscle tissue.

5) Importantly, from Figure 3F, the main conclusion that "the addition of amino acid building blocks is sufficient to rescue a TRIM32-mediated loss of muscle mass." is not warranted.The addition of yeast extract or amino acids does not 'rescue' the muscle defect of tn mutants, i.e. it does not return the phenotype to wildtype levels. It simply rescues the additional defect caused by not feeding these animals any protein (agar only). This does not mean much. It is analogous to not feeding tn mutants any water and seeing that they dehydrate. Then giving them water again would return them to the original tn mutant phenotype, and thereby concluding that water is sufficient to rescue the TRIM32-mediated phenotype.To prove that amino acid deficiency is causing the muscle atrophy phenotype, one would need to see that feeding additional amino acids (on top of the normal food) to tn mutants causes their muscle size to become more like wildtype.

This entire section has been rewritten.

6) Could it be the other way around – that a muscle defect is leading to an eating defect (which requires continuous muscle movement) and as a consequence, less intake of sugar and protein by the larvae, leading to the metabolic phenotypes? This can be resolved by:– quantifying food intake by tn mutant larvae;– checking the cell-autonomy of the metabolic defects;– testing in cell culture whether TRIM32 knockdown leads to the same glycolytic defects.

This is a really great point and has now been addressed. Please see previous responses (Essential revisions #1; reviewer #1, point #1; reviewer #2, point #3)

7) Figure 4—figure supplement 1: it is difficult to judge whether apoptosis rates account for a difference in tissue size by just quantifying apoptosis, because this only yields a snapshot of apoptosis at that moment, whereas tissue size integrates apoptosis rates over time. So a change in apoptosis may appear mild at any one point, but result in significant tissue size differences over time. The way to prove this functionally is to block apoptosis, e.g. with p35. Alternatively, if the authors want to conclude that cell size in tn brains is smaller than in control brains, a direct quantification of cell size would be better.

We have now measured the area of cells in the lobes of the developing larval brain. Although quite variable in size depending on their cell lineage and/or the timing of cell division, there is a clear overall reduction in cell area upon loss of TRIM32 (Figure 4—figure supplement 1J-M).

8) The authors write "Wing discs are not inherently glycolytic and do not endogenously express LDH (Figure 4E). Accordingly, the overall area or volume of the disc was not affected by loss of TRIM32"This is not correct. Endogenous LDH (called ImpL3 in Drosophila) is expressed in wing discs, and can be detected by Q-RT-PCR or by in situ hybridization on endogenous transcript.Hence, the fact that wing disc size is not decreased in TRIM32 mutants cannot be explained with this justification and raises the question whether indeed the metabolic mechanism the authors propose is correct.

Thank you for pointing this out. We simply meant that wing discs are not considered a tissue with enhanced glycolytic activity and that LDH-GFP is not enriched in *WT* wing discs [consistent with published data that endogenous LDH activity is not readily detectable in wing and eye discs (Wang et al., 2016)]. We have corrected this statement to read, ‘Unlike muscle or brain tissue, neither LDH activity (Wang et al., 2016) nor LDH-GFP expression (Figure 5A) were detectable in control wing discs, suggesting that this tissue does not exhibit elevated glycolytic activity.’

9) Along the same lines, there is no quantification of glycolytic flux rates in the wing discs – neither that PVR expression causes them to increase, nor that the tn mutation causes them to drop again. The phenotype shown in Figure 4H simply says there is some genetic interaction, but this could be happening at any level.We now include FILM data (Figure 5G) that demonstrate differences in the glycolytic profile between PVR tumors grown in *WT* or *tn-/-* larvae. Although variable among individual discs, there is a trend of elevated glycolytic flux in PVR-induced tumors that is significantly reduced when these tumors are grown in *tn-/-* larvae. Note that PER analysis is not possible with *tn-/-; dpp>Pvr^act^* discs since the isolated tumors are small and does not give enough protein for normalization. It was not trivial to isolate *WT* or *tn-/-* wing discs with proper normalization due to small size of the tissue (Figure 5—figure supplement 2D).Reviewer #3:[…]1) Proton efflux rate (PER), as shown with brain tissues in Figure 4D, provides a reasonable assessment of glycolytic flux in isolated tissues. In contrast, extracellular acidification rate (ECAR) is also influenced by CO_2_ produced during respiration; thus it wouldn't provide an accurate estimation of glycolytic flux. To clearly demonstrate Drosophila TRIM32 regulate glycolysis in larval muscle, it is critical to compare PER in wild-type and tn muscles.

We agree that this experiment is critical. We now include PER data for dissected *WT* and *tn-/-* muscle tissue (Figure 2I). Since the PER assay reports acidification of the media due to glycolysis and not resulting from TCA cycle metabolism, our results clearly show that loss of TRIM32 reduces glycolytic flux in muscle tissue.

2) Only one tn mutant allele is used throughout the study. How can the authors rule out the possibility that the phenotypes observed with the tn allele is not due to a background mutation? One way to resolve this issue is to show that the key phenotypes, such as a reduction in PER, can be reversed by having a genomic rescue transgene or tissue specific rescue using GAL4/UAS system. Alternatively, it would be helpful to show that similar phenotypes can be seen with different tn alleles.

We now have this data. Please see previous responses (Essential revisions #2; reviewer #1, point #2).

3) Considering the hypothesis that TRIM32 is required for maintaining glycolytic flux, I would suggest to move the results in Figure 2—figure supplement 2C-D to the main figures. Note that the results shown in Figure 2C-G are not new; additionally, similar results are shown in Figure 3B-F. Thus, Figure 2C-G can be moved to the supplementary figures.

We appreciate this suggestion. In our revised version, we moved Western blots that show TRIM32 regulates Ald and Pglym protein levels to Figure 1E and G. We chose not to move the muscle pictures in Figure 2C-G to supplementary data as we want to clearly establish the decrease in muscle cell size during larval development (L2 to L3 stages).

[Editors' note: further revisions were suggested prior to acceptance, as described below.]

The manuscript has been improved but there are some remaining issues that need to be addressed before acceptance, as outlined below:1) As suggested by reviewer #1 comment 2, reviewer #2 comment 1-3, in vivo interactions between TRIM32 and Ald/Pglym are critical to prove the hypothesis, and therefore, require precise descriptions of experiments and results.

We have now included this data in a new Figure 2 and Figure 2—figure supplement 1. We observe a higher molecular weight form of Ald and Pglym upon immunoprecipitation of TRIM32 from L3 larval lysates. Note that we can visualize the modified forms of these proteins using antibodies generated against *Drosophila* Ald or Pglym (Figure 2C,D) as well as antibodies against human ALD or PGAM1 that cross-react with the fly protein (Figure 2—figure supplement 1D, E). Experimental descriptions, methods, and results are included in the modified manuscript. Please also see explanations in response to reviewers #1 and #2 below.

2) Reviewer #3 comment 2: ERR-mediated restoration experiments need better descriptions and if possible, additional data could be added.

Data showing that ERR expression restores glycolytic protein levels was included in our previous revision in a supplementary figure. This data has now been moved to a new Figure 4 where we show that ERR, which transcriptionally induces glycolytic gene expression, also restores muscle mass and muscle contraction upon loss of TRIM32. Please see detailed explanations in response to reviewer #3 below.

3) Other comments can be immediately addressed or edited.

Other comments have been addressed in the manuscript and are detailed below.

Reviewer #1:Authors have significantly improved the manuscript and adequately addressed critiques raised by reviewers. There are some points below to be changed for better readability.1) In the line "Importantly, protein levels of Ald.… ERR in tn-/- muscles, indicating that~", figure citation is missing. It could be Figure 2—figure supplement 2F-G.

Thank you for catching this. Our failure to add this citation explains why reviewer #3 did not notice this data (#2 above in Essential revisions). This data has now been moved to a new Figure 4I and J.

2) As suggested in the major revision 3, showing the in vivo interactions between TRIM32 and Ald/Pglym is essential for the study. Therefore, Author response image 1 needs to be moved to the main figure with additional explanations and discussions.

As discussed above in Essential revision #1, we have now included this data a new Figure 2 and Figure 2—figure supplement 1.

3) Given that tn mutants exhibit reduced mouth hook contraction, the growth of tn mutants could be overall retarded while tissue-specific knockdown of tn showed no difference. Having the possibility of slow growth, showing the small-sized muscle or brain of tn mutants in main Figures 3-4 could be misleading. It will be more adequate to show the small size phenotypes with tissue-specific RNAi backgrounds.

We have now moved tissue-specific data that was previously in supplementary data to main Figure 4. Specifically, knockdown of TRIM32 in muscle reduces muscle size (Figure 4A-D), but does not alter growth of the larval brain (Figure 6E-G).

Reviewer #2:The manuscript is significantly improved. For instance, the authors do a good job of showing the tissue-autonomous nature of the TRIM32 phenotypes.Some issues which can be fixed without any additional experiments:1) The authors write in their rebuttal "Identification of the TRIM32-Ald or TRIM32-Pglym biochemical interaction was performed in vivo using larval lysates (Figure 1C; Figure 1—figure supplement 2A…". If I understand correctly, in Figure 1, a truncated version of TRIM32 was fused to GST, expressed recombinantly in bacteria, and purified. This was then incubated in a test tube with larval lysates. How can this be called "in vivo"? Some people call cells in culture "in vivo" and some people mean a live organism by "in vivo", but I do not think anyone considers a lysate in a test tube to be "in vivo"? If I misunderstood the experimental setup, it needs to be described better in the manuscript.

The reviewer is correct and we now include in vivo data showing that immunoprecipitation of TRIM32 co-purifies with Ald and Pglym (new Figure 2C, D).

2) Showing that full-length TRIM32 binds Ald and Pglym in vivo is critical for this story. I believe the results shown in Author response image 1 should be put into the manuscript.

As stated above, we moved this data to Figure 2.

3) Regarding the fact that the co-IPed Ald and Pglym in Author response image 1 are running at a different molecular weight compared to Ald and Pglym in the input can either be interpreted as the authors write by post-translational modification of the pool of Ald and Pglym that interact with TRIM32, or it might simply be another contaminating protein that cross-reacts with the Ald and Pglym antibodies. Can this be distinguished? If it is a sub-population of the total Ald/Pglym, it should also be visible in the total lysates. Also, most post-translational modifications are readily lost in IP buffer and Laemmli (e.g. ubiquitination or phosphorylation), which would cause it to run at the 'normal' size. This should be discussed.

Great point. We have verified the higher molecular weight bands corresponding to Ald or Pglym using additional antibodies raised against human ALD or PGAM1. Due to the high conservation of glycolytic enzymes, these antibodies cross-react with fly proteins. Note that IPs were carried out with guinea pig antibodies and Western blots were performed with antibodies raised in rabbit, ruling out heavy or light chain reactivity. We also can see a small proportion of this modified band when we overexpose concentrated lysates and have included discussion of these results.

4) The authors write "Surprisingly, cellular glucose uptake… was normal in tn-/- larval brains and wing discs… demonstrating that glucose is not a limiting substrate for glycolysis in these isolated tissues."I believe this statement is not correct. How does this show glucose is not a limiting substrate for glycolysis in brains and wing discs? Or do the authors mean specifically in tn mutants?

Yes, we meant specifically in *tn* mutants. This has been corrected.

Reviewer #3:The authors have appropriately addressed most of my comments. I have two additional comments:1) The authors showed that Ald and Pglym protein levels were reduced in tn mutant. However, the authors didn't show that TRIM32 stabilized these proteins. Thus, I suggest to revise the following sentences:

This data was present in our submitted revision (Figure 2—figure supplement 2F-G). To make this point more clearly, we have now moved this data to a new Figure 4I, J. Hence, we have chosen to keep the following sentences as written.

"Here we provide a novel mechanism for TRIM32 in cell growth. Our data show that TRIM32 promotes glucose metabolism through the stabilization of glycolytic enzyme levels.""Taken together, these data highlight a unique role for TRIM32 in the stabilization of glycolytic enzyme levels.""Importantly, protein levels of Ald and Pglym were stabilized upon expression of ERR in tn-/- muscles, indicating that restoration of glycolytic protein levels is sufficient to recover TRIM32-mediated growth defects."2) Although the authors claimed that ERR overexpression stabilized Ald and Pglym levels, the results were not shown. Thus, the authors should test whether ERR overexpression could restore Ald and Pglym levels in tn mutant. Otherwise, the result regarding the ERR overexpression in muscle shown in Figure 2M is better to be removed. Additionally, given that ERR can impact on multiple targets, "restoration of glycolytic protein levels is sufficient to recover TRIM32-mediated growth defects." is an overstatement.

As mentioned previously, this data was present in our submitted revision (Figure 2—figure supplement 2F-G). To make it more visible, we have now moved this data to a new Figure 4I, J. Not only do we show that ERR can stabilize protein levels in *tn-/-,* but it can also restore muscle size and muscle contraction.